# Bovine Pancreatic RNase A: An Insight into the Mechanism of Antitumor Activity In Vitro and In Vivo

**DOI:** 10.3390/pharmaceutics14061173

**Published:** 2022-05-30

**Authors:** Islam Saber Ead Mohamed, Aleksandra V. Sen’kova, Oleg V. Markov, Andrey V. Markov, Innokenty A. Savin, Marina A. Zenkova, Nadezhda L. Mironova

**Affiliations:** 1Institute of Chemical Biology and Fundamental Medicine, Siberian Branch of the Russian Academy of Sciences, 630090 Novosibirsk, Russia; sabermohamedm28@gmail.com (I.S.E.M.); senkova_av@niboch.nsc.ru (A.V.S.); markov_ov@niboch.nsc.ru (O.V.M.); markov_av@niboch.nsc.ru (A.V.M.); savin_ia@niboch.nsc.ru (I.A.S.); marzen@niboch.nsc.ru (M.A.Z.); 2Faculty of Natural Sciences, Novosibirsk State University, 630090 Novosibirsk, Russia; 3Faculty of Science, Al-Azhar University, Assiut 71524, Egypt

**Keywords:** RNase A, ribonuclease inhibitor, antitumor activity, intracellular localization, Ku70/Ku80, miRNA, EMT markers

## Abstract

In this investigation, we extensively studied the mechanism of antitumor activity of bovine pancreatic RNase A. Using confocal microscopy, we show that after RNase A penetration into HeLa and B16 cells, a part of the enzyme remains unbound with the ribonuclease inhibitor (RI), resulting in the decrease in cytosolic RNAs in both types of cells and rRNAs in the nucleoli of HeLa cells. Molecular docking indicates the ability of RNase A to form a complex with Ku70/Ku80 heterodimer, and microscopy data confirm its localization mostly inside the nucleus, which may underlie the mechanism of RNase A penetration into cells and its intracellular traffic. RNase A reduced migration and invasion of tumor cells in vitro. In vivo, in the metastatic model of melanoma, RNase A suppressed metastases in the lungs and changed the expression of EMT markers in the tissue adjacent to metastatic foci; this increased Cdh1 and decreased Tjp1, Fn and Vim, disrupting the favorable tumor microenvironment. A similar pattern was observed for all genes except for Fn in metastatic foci, indicating a decrease in the invasive potential of tumor cells. Bioinformatic analysis of RNase-A-susceptible miRNAs and their regulatory networks showed that the main processes modulated by RNase A in the tumor microenvironment are the regulation of cell adhesion and junction, cell cycle regulation and pathways associated with EMT and tumor progression.

## 1. Introduction

The antitumor potential of exogenous nucleases has been studied for about 65 years. Among the known ribonucleases exhibiting antitumor activity, most notable are onconase (ribonuclease from frog oocytes) [1], BS-RNase (bovine testicular RNase) [2,3], fungal RNases (α-sarcin, mitogillin and restrictocin) [4] and microbial RNases belonging to the RNase A superfamily [5,6,7].

Bovine pancreas ribonuclease A (RNase A) is a small protein (124 amino acids, 13.7 kDa) and has the highest catalytic activity among the proteins of its superfamily [8]. RNase A is the first ribonuclease being studied in vitro for its cytotoxicity and in vivo for antitumor potential; however, controversial results have been obtained [9,10,11,12,13,14]. In some studies, RNase A was shown to inhibit tumor growth in mice and rats at doses up to 1000 mg/kg, while in others the absence of both cytotoxic and antitumor effects was observed, and the lack of antitumor effects was explained by the tight binding of RNase A to intracellular ribonuclease inhibitor (RI) [2,8,11,12,15,16,17]. Nevertheless, despite the strong binding to RI, dimeric forms of RNase A were shown to exhibit significant antitumor effects [18]; thus, the question about the antitumor potential of RNase A was raised again.

For the past 10 years, data were accumulated demonstrating in different murine tumor models the high antitumor and antimetastatic potential of bovine pancreas RNase A used at doses not exceeding 10 μg/kg [19,20]. The discovery of extracellular circulating RNAs and the elucidation of their role in tumor progression and metastases spreading [21,22] suggested that these RNAs together with intracellular RNAs could be the molecular targets for RNase A. There have been attempts to identify molecular targets of RNase A in the tumor tissue and blood of tumor-bearing mice (with the example of Lewis lung carcinoma) [23]. It was revealed that possible mechanisms underlying the antitumor and antimetastatic activity of RNase A include the degradation of circulating RNAs, including microRNAs (miRNAs), and alterations in miRNA patterns in tumor tissue [23]. Moreover, it was shown that the ribonuclease activity of RNase A plays a crucial role in these events [23]. On the transcriptome level, RNase A caused the reorganization of intracellular networks of tumor cells, providing an enhancement of energy cascade activity, the inhibition of cell proliferation and dissemination processes and the partial depletion of signaling pathways exhibiting tumor-promoting activity [24].

Thus, there are separate elements of the mosaic from which it is possible to create a picture of the antitumor and antimetastatic action of RNase A. However, accumulating a large amount of data on the influence of RNase A on tumor progression in vivo, only a few studies have been devoted to what happens in tumor cells in vitro. Moreover, the question of the effect of RNase A on intracellular RNAs still remains open.

Ku is one of the cellular proteins executing multiple functions whose intracellular concentration is amounted to 1.5 μM [25]. Ku is a complex composed of two tightly bound subunits called Ku70 and Ku80 [26] localized both in the cytosol and the nucleus. Ku has long been considered as a nuclear protein playing a role in DNA repair [27], but there is now some evidence of Ku functions in cytosol and on the outer membrane: cytosolic DNA sensing and subsequent innate immune response activation [28], apoptosis regulation, mitosis [29], hypoxia, metabolism and inflammatory response [30]. Ku can recognize RNA hairpins, although less effectively than dsDNA [31], so it is considered an RNA-binding protein.

Recent reports demonstrate that some mammalian cells are using the DNA repair protein Ku in a membrane-associated form for interaction with their microenvironment composed of other cellular components and an extracellular matrix (ECM) [32]. The functions of the membrane-associated Ku are not restricted to cell–cell interaction but also include cell–matrix interaction. Additional results, although mainly descriptive, raised the idea that Ku might also play a role in cell migration and invasion. On the other hand, several studies have revealed the cytoplasmic and the cell-surface localization of Ku proteins in a variety of tumor cells, including leukemia, multiple myeloma and solid tumor cell lines [33]. Taking into account the high abundance of the Ku protein in the tumor cells and its participation in cell contacts and invasion, it seemed interesting to study the possible interaction of the Ku protein with exogenous RNase A in tumor cells.

Here, using two tumor cell lines, mouse melanoma B16 and human epidermoid carcinoma HeLa, we studied the penetration of RNase A into the cells and its intracellular interaction with RI and with Ku70/Ku80 heterodimer. We evaluated the effects of RNase A on the migration activity, motility and pro-metastatic properties (adhesion, invasion and colony formation) of B16 and HeLa cells in vitro and on metastasis spread in the melanoma B16 model in vivo. The alteration of miRNAs of both oncomirs and oncosuppressors as well as the expression of markers associated with the epithelial–mesenchymal transition (EMT) under the action of RNase A were studied in blood, metastatic foci and adjacent lung tissue (modeling tumor environment) in the context of inhibiting tumor progression in vivo. Additionally, regulatory networks of RNase-A-susceptible miRNAs were reconstructed and analyzed.

## 2. Materials and Methods

### 2.1. Cell Cultures

Murine melanoma B16 cell line was purchased from N.N. Blokhin Cancer Research Centre (Moscow, Russia). HeLa cells were obtained from the collection of ICBFM SB RAS (Novosibirsk, Russia). B16 and HeLa cells were maintained on Dulbecco’s Modified Eagle Medium (DMEM) containing 10% fetal bovine serum (FBS) and 1% antibiotic–antimycotic solution (10 mg/mL streptomycin, 10,000 U/mL penicillin, 25 μg/mL amphotericin (MP Biomedicals, Irvine, CA, USA)) (hereafter, complete medium) at 37 °C in a humidified atmosphere with 5% CO_2_ (hereafter, standard conditions).

### 2.2. Mice

Male 10–14-week-old C57Bl/6 (hereafter, C57Bl) mice were obtained from the vivarium of ICBFM SB RAS (Novosibirsk, Russia). Mice were housed in plastic cages (10 animals per cage) under normal daylight conditions. Water and food were provided ad libitum. All animal procedures were carried out in strict accordance with the approved protocol and recommendations for the proper use and care of laboratory animals (ECC Directive 2010/63/EU). The experimental protocols were approved by the Committee on the Ethics of Animal Experiments with the Institute of Cytology and Genetics SB RAS (ethical approval no. 50 dated 23 May 2019), and all efforts were made to minimize suffering.

### 2.3. Scratch Assay

Cells were grown in 24-well plates in 500 μL of medium per well until confluence was reached (1.0 × 10^6^ cell/well). The culture medium was replaced by medium supplemented with RNase A (2.5–10 μg/mL). A wound was made by scratching the cells with a 200 μL pipette tip followed by 72 h incubation. The scratched monolayer was photographed at 0, 24, 48 and 72 h after seeding using a Zeiss Primo Vert microscope (Zeiss, Germany). Cell migration was assessed by measuring gap sizes at multiple fields using ImageJ 1.50f. The migration width was estimated as the difference between scratch width filled with cells after 48 h and the initial scratch width. The rate of cell migration was assessed by calculating the degree of scratch overgrowth using the equation:υ = (1 − Χ) × 100%, 
where Χ is the ratio of the cell-free area of the scratch to the initial area of the scratch [34].

### 2.4. Influence of RNase A on the Migration Activity, Motility, Adhesion, Invasion and Colony Formation of B16 and HeLa Cells

For the cell motility assay, B16 and HeLa cells were seeded in FBS- and antibiotic-free DMEM to the upper chamber of a 16-well CIM plate (pore size 8 μm) at a density of 2 × 10^4^ cells per well. RNase A was added in the upper chamber at concentrations of 2.5, 5 and 10 µg/mL. To stimulate cell motility, DMEM with 10% FBS and chemoattractant was placed in the lower chamber of the CIM plate. The cell index (electrical impedance) was monitored in real time using the xCELLigence DP instrument (ACEA Biosciences, San Diego, CA, USA) for 48 h under standard conditions.

For the invasion assay, B16 and HeLa cells were seeded in FBS- and antibiotic-free DMEM in the upper chamber of a 16-well CIM plate (pore size 8 μm) precoated with Matrigel at a density of 2 × 10^4^ cells. DMEM with 10% FBS was placed in the lower chamber of the CIM plate. RNase A was added to the upper chamber at concentrations of 2.5, 5 and 10 µg/mL, and cells were incubated for 48 h under standard conditions. Analysis of invasion was monitored in real time using the xCELLigence DP instrument [35]. Cell index was measured every 1 h for up to 48 h with the RTCA software (ACEA Biosciences, San Diego, CA, USA).

For the cell adhesion assay, B16 and HeLa cells were seeded in antibiotic-free DMEM in a 24-well plate at a density of 0.5 × 10^6^ cells/well and incubated in the presence of RNase A at concentrations of 2.5, 5 and 10 µg/mL for 24 h at 37 °C and 5% CO_2_. Cells were trypsinized and resuspended in 1 mL of DMEM. Then, 100 µL of cell suspension was added per well of a 96-well plate and allowed to adhere for 1 h under standard conditions. After incubation, non-adherent cells were removed by washing. Next, 10 µL of MTT solution in DMEM was added to the cells (final concentration: 0.5 mg/mL), and the plate was incubated for 3 h under standard conditions. Then, the MTT-containing medium was discarded, and formazan crystals were solubilized with 100 µL of DMSO. The absorbance of each well was read at test and reference wavelengths of 570 and 620 nm, respectively, on a Multiscan RC plate reader (Thermo LabSystems, Helsinki, Finland).

Colony formation assay was performed using a methodology adapted for the 96-well plate [36]. B16 and HeLa cells were seeded in a 96-well plate in antibiotic-free DMEM supplemented with 10% FBS at a density of 200 cells/well. RNase A was added at concentration range of 0.625–10 µg/mL, and cells were incubated for 8 and 14 days under standard conditions. Then, cell colonies were fixed with 4% paraformaldehyde, stained with crystal violet dye (0.1% *w*/*v*) and photographed using iBright™ CL1500 Imaging Systems (Thermo Fisher Scientific, Waltham, MA, USA). The percentage of the area covered by cell colonies was calculated using a ColonyArea ImageJ plugin (National Institutes of Health, Bethesda, MD, USA).

### 2.5. Conjugation of RNase A with Biotin

RNase A (from bovine pancreas, lyophilized, 13,700 g/M, Sigma-Aldrich, Darmstadt, Germany) was conjugated with biotin using the reagent EZ-Link Sulfo-NHS-LC-Biotin (Thermo Scientific) in accordance with the manufacturer’s protocol. Briefly, 3 mg of RNase A diluted in 900 μL of 0.1 M NaHCO_3_, pH 8.3, was mixed with 2.44 mg of EZ-Link Sulfo-NHS-LC-Biotin (Thermo Fisher Scientific, Waltham, MA, USA) diluted in 100 µL of milliQ water and incubated at room temperature for 30 min. The RNase A–biotin conjugate was purified by centrifugation on Amicon Ultra 10K MWCO (Millipore, Burlington, MA, USA) columns according to the manufacturer’s protocol. The degree of biotin conjugation with RNase A was evaluated using Western blot analysis. RNase A after conjugation with biotin retained its ribonuclease activity, which was confirmed by the cleavage of miR-21-3′-FAM.

### 2.6. RNA Visualization by Fluorescence Microscopy

HeLa and B16 cells in complete DMEM were seeded on cover slips placed in a 24-well plate at densities of 0.5 × 10^5^ and 1 × 10^5^ cells/well, respectively, and incubated under standard conditions overnight. On the next day, culture medium was replaced with fresh DMEM supplemented with 1, 5 and 20 µM of RNase A–biotin conjugate, and cells were cultured for 4 and 48 h under standard conditions. Cells were washed twice with PBS supplemented with 2% FBS, fixed by cold methanol at −20 °C for 10 min and washed with PBS for 5 min. Cells were stained with 0.5 μM SYTO RNASelect (Thermo Fisher Scientific, Waltham, MA, USA) for 30 min at room temperature in the dark [37]. Finally, cells were washed three times by PBS and mounted on slides ProLong™ Glass Antifade Mountant with NucBlue™ Stain (Thermo Fisher Scientific, Waltham, MA, USA). Mounted samples were allowed to cure on a flat, dry surface for 18–24 h at room temperature in the dark and analyzed using confocal microscopy: SYTO, ex. 490/nm, em. 530 nm.

The fluorescence intensities of intracellular RNA were measured in the sample images obtained with confocal fluorescent microscopy on LSM710 (Zeiss, Zeiss, Oberkochen, Germany) by using ImageJ software [38].

Fluorescence intensity was calculated using the equation:Fluorescence intensity, % = (RFU_experimental group_/RFU_control group_) × 100%, 
where RFU_control group_ is the fluorescence intensity of non-treated cells, and RFU_experimental group_ is the fluorescence intensity of cells treated with RNase A.

### 2.7. Analysis of Accumulation and Intracellular Localization of RNase A–Biotin Conjugate in HeLa and B16 Cells by Confocal Microscopy

HeLa and B16 cells were seeded in complete DMEM on coverslips in 24-well plates at a density of 0.4 × 10^5^ cells/well and incubated under standard conditions overnight. The culture medium was replaced with FBS- and antibiotic-free DMEM supplemented with 20 µM of RNase A–biotin conjugate, and the cells were incubated under standard conditions for 4 h.

After incubation, coverslips with cells were washed with PBS and fixed in 4% formaldehyde (Sigma-Aldrich, St. Louis, MO, USA) in PBS for 20 min at 37 °C, then washed two times with PBS. Cells were permeabilized by incubation in PBS containing 0.1% Tween 20 and 0.5% BSA for 10 min at room temperature. After that, cells were washed twice with PBS and incubated with blocking buffer (PBS containing 0.05% Tween 20 and 1% BSA) for 30 min at room temperature. Cells were stained with anti-RI rabbit polyclonal antibodies (RAI, Ab217132, Abcam, Cambridge, MA, USA) (1:300) and incubated for 1 h at room temperature in the dark. Then, cells were washed once with PBS and incubated with goat anti-rabbit antibodies conjugated with Alexa Fluor 488 (ab150077, Abcam, Cambridge, MA, USA) (1:1000) and eBioscienceTM Streptavidin APC (Thermo Fisher Scientific, Waltham, MA, USA) (0.2 mg/mL) for 30 min at room temperature in the dark. Finally, cells were washed three times by PBS and mounted on slides in a drop of ProLong™ Glass Antifade Mountant with NucBlue™ Stain (Thermo Fisher Scientific, Waltham, MA, USA). Mounted samples were allowed to cure on a flat, dry surface for 18–24 h at room temperature in the dark.

Intracellular localizations of RNase A and RI were assessed with confocal fluorescent microscopy on LSM710 (Zeiss, Oberkochen, Germany) using a plan-apochromat 63x/1.40 Oil DIC M27 objective, and ZEN software (Zeiss, Oberkochen, Germany) was used for analysis. Confocal analysis was performed in three channels (blue, green, red): Nucblue, ex. 359 nm, em. 457 nm; Alexa Fluor 488, ex. 490/nm, em. 525 nm; APC, ex. 650 nm, em. 660 nm. Fluorescence in the blue channel corresponded to NucBlue (nuclei staining), the green channel corresponded to RI stained with Alexa Fluor 488–antibodies and the red channel corresponded to RNase A–biotin stained with streptavidin-APC.

### 2.8. Analysis of RNase A–Biotin Conjugate Accumulation into B16 and HeLa Cells by Flow Cytometry

HeLa and B16 cells were seeded in complete DMEM in 24-well plates at a density of 0.4 × 10^5^ cells/well and incubated under standard conditions overnight. The culture medium was replaced with FBS- and antibiotic-free DMEM supplemented with 1–20 µM of RNase A–biotin conjugate, and the cells were incubated under standard conditions for 4 h. Cells were prepared for streptavidin-APC staining as described in Section 2.7. Cells were stained with eBioscienceTM Streptavidin APC (0.2 mg/mL) (Thermo Fisher Scientific, Waltham, MA, USA) and incubated for 30 min at room temperature in the dark. Cells were washed with PBS and analyzed by flow cytometry using a NovoCyte (ACEA Biosciences, San Diego, CA, USA). Data were processed with NovoExpress software (ACEA Biosciences, San Diego, CA, USA) as described in [39].

The mean fluorescence intensity of cells was measured in relative fluorescence units (RFU). All experiments were run in triplicate for statistical analysis; the standard deviation of the means did not exceed 5%. The efficacy of RNase A penetration was characterized by a percentage of APC-positive cells and mean fluorescence intensity in the sample. The percentage of APC-positive cells in the analyzed samples was calculated using the equation:APC^+^(%) = APC^+^_sample_ (%) − APC^+^_control_ (%), 
where APC^+^_sample_ (%) is the percentage of fluorescence-positive cells in the analyzed sample and APC^+^_control_ (%) is the percentage of fluorescence-positive cells in the negative control (autofluorescence).

Cells incubated in the absence of RNase A–biotin conjugate were used as a negative control.

### 2.9. Co-Localization of RNase A–Biotin Conjugate with Ku70/Ku80 in B16 and HeLa Cells

HeLa and B16 cells were seeded in complete DMEM on glass cover slips in 24-well plates at densities of 0.4 × 10^4^ and 0.8 × 10^4^ cells/well, respectively, and incubated under standard conditions overnight. The culture medium was replaced with a serum- and antibiotic-free DMEM medium supplemented with 20 µM of RNase A–biotin conjugate, and the cells were incubated under standard conditions for 4 h. Cells were prepared for antibody staining as described in Section 2.7.

Cells were stained with Ku70/Ku80 Monoclonal Antibody (162) (MA1-21818, Thermo Fisher Scientific, Waltham, MA, USA) (1:100) and incubated for 1 h at room temperature in the dark. Then, cells were washed once with PBS and incubated with goat anti-mouse IgG (H + L) highly cross-adsorbed secondary antibody, Alexa Fluor Plus 488 (A32723, Abcam, USA) (1:500) and eBioscienceTM Streptavidin APC (Thermo Fisher Scientific, Waltham, MA, USA) (0.2 mg/mL) for 30 min under the same conditions. Finally, cells were prepared for confocal microscopy and analyzed as described in Section 2.7.

Confocal microscopy analysis was performed in three channels (blue, green, red). Fluorescence in the blue channel corresponded to NucBlue (nuclei staining), the green channel corresponded to Ku70/Ku80 stained with Alexa Fluor 488 antibodies and the red channel corresponded to RNase A–biotin conjugate stained with streptavidin-APC: Nucblue, ex. 359 nm, em. 457 nm; Alexa Fluor 488, ex. 490/nm, em. 525 nm; APC, ex. 650 nm, em. 660 nm.

### 2.10. B16 Implantation and Design of the Animal Experiment

The B16 metastatic model was induced in mice by intravenous (i.v.) injection of B16 melanoma cells (0.1 mL, 10^6^ cells/mL) into the lateral tail vein of C57Bl mice. On day 4 after tumor implantation, mice with B16 were assigned to three groups (*n* = 10 per group): (1) control, received i.m. saline buffer (0.1 mL); (2) and (3) received i.m. RNase A at doses of 0.7 or 7.0 µg/kg (0.1 mL), respectively. Injections were carried out for two weeks using the 5 + 2 scheme (one injection per day for 5 days, a 2-day pause). The total number of injections was ten. On day 15 after tumor implantation, 1 h after the last injection of RNase A, blood samples were taken from the retro-orbital sinus. Animals were sacrificed under light ether anesthesia, the lung samples (pieces with metastasis and pieces of adjacent tissue) were collected and fixed in 10% neutral buffered formalin (pH 7.0; BioVitrum, St. Petersburg, Russia) for subsequent pathomorphological analysis. Part of the samples was used for RNA isolation. The number of surface metastases in the lungs was determined using a binocular microscope.

### 2.11. Sample Processing and RNA Extraction

Blood serum was prepared from the whole blood of animals treated with saline buffer or RNase A by clot formation at 37 °C for 30 min and at 4 °C overnight, followed by clot discard and centrifugation (4000 rpm, 4 °C, 20 min) to remove cell debris. Lung pieces were homogenized.

B16 and HeLa cells were seeded in complete DMEM in 12-well plates at a density of 1.5 × 10^6^ cells/well, RNase A (5 μg/mL) was added, and cells were incubated for 48 h under standard conditions and collected for total RNA isolation.

exRNA from serum samples and total RNA from B16 and HeLa cells and homogenates of lung tissue were extracted using TRIzol reagent (Invitrogen, Thermo Fisher Scientific, Waltham, MA, USA) according to the manufacturer’s protocol. The RNA concentrations in the samples were measured by absorbance at 260 and 280 nm using a NanoDrop™ OneC Spectrophotometer (Thermo Fisher Scientific, Waltham, MA, USA).

### 2.12. RT-qPCR

The levels of miRNAs in blood serum and expression of miRNAs in B16 and HeLa cells and lung tissues were analyzed using stem loop RT-qPCR technology [40,41]. Measurement of the expression of EMT marker genes Cdh1, Vim, Fn and Tjp was performed using RT-qPCR.

cDNA synthesis was performed in reaction mixture (40 µL) containing 5 µg of total RNA, 5× RT buffer mix, 100 units of M-MuLV-RH reverse transcriptase (Biolabmix, Russia), 0.05 µM of miRNA specific stem-loop primers or 0.05 µM of hexaprimer (Appendix A). The RT reaction was as follows: 16 °C, 30 min; 30 °C, 30 s; 42 °C, 30 s; 40 cycles; and a final reverse transcriptase inactivation at 85°C for 5 min.

qPCR was carried out in a total volume of 20 µL using 2× BioMaster HS-qPCR SYBR Blue (Biolabmix, Novosibirsk, Russia), 0.05 µM of forward miRNA-specific primers and universal reverse primer or 0.05 µM of forward and reverse specific primers (Appendix A). The PCR conditions for miRNAs were as follows: 95 °C, 5 min; 95 °C, 15 s; 58 °C, 15 s, 40 cycles; 72 °C, 30 s; 75 °C, 15 s; followed by a melting point determination. The PCR conditions for EMT markers were as follows: 95 °C, 5 min; 95 °C, 10 s; 51 °C, 30 s, 40 cycles; 72 °C, 30 s; followed by a melting point determination.

The obtained PCR data were analyzed using CFX Maestro Software version 1.0 (Bio-Rad Laboratories Inc., Hercules, CA, USA). For each sample, the threshold cycle (Ct) was determined. Quantitative assessments of the level of transcripts representation and relative miRNA expression in tumor cells and tumor tissue were performed by comparing the Ct values for miRNA and the reference U6 snRNA. The concentration of serum-derived miRNAs was normalized to the serum volume. For EMT genes, the reference gene HPRT was used.

### 2.13. Histology and Immunohistochemistry

For histological study, the lung specimens were fixed in 10% neutral buffered formalin (BioVitrum, St. Petersburg, Russia), dehydrated in ascending ethanols and xylols and embedded in HISTOMIX paraffin (BioVitrum, St. Petersburg, Russia). The paraffin sections (5 μm) were sliced on microtome Microm HM 355S (Thermo Fisher Scientific, Waltham, MA, USA) and stained with hematoxylin and eosin. Inhibition of metastases development was assessed by morphometry using the metastasis inhibition index (MII), calculated as MII = [(mean metastasis area_control_ − mean metastasis area_experiment_)/mean metastasis area_control_] × 100%. The MII of the control group was taken as 0%, and the MII corresponding to 100% reflected the absence of metastases.

For the immunohistochemical study, the lung sections (3–4 μm) were deparaffinized and rehydrated. Antigen retrieval was carried out after exposure in a microwave oven at 700 W. The samples were incubated with the anti-E-cadherin (ab76055, Abcam, Cambridge, MA, USA) or anti-Vimentin (ab92547, Abcam, Cambridge, MA, USA) specific primary antibodies according to the manufacturer’s protocol. Then, the sections were incubated with secondary Alexa Fluor^®^ 488-conjugated antibodies and embedded in Fluoromount-GTM Mounting Medium (Invitrogen, Thermo Fisher Scientific, Waltham, MA, USA). All the images were examined and scanned using Axiostar Plus microscope equipped with fluorescent lamp HBO 50W/AC L1 (Osram, Munich, Germany) and Axiocam MRc5 digital camera (Zeiss, Oberkochen, Germany) at magnifications of ×100 (hematoxylin and eosin images) and ×200 (fluorescence-based immunohistochemistry).

### 2.14. Molecular Modeling of RNase A Interaction with Ku70/Ku80

The crystal structures of RNase A (1FS3) and heterodimer Ku70/Ku80 (1JEQ) obtained from the Protein Data Bank database were freed from water molecules using BIOVIA Discovery Studio, and their ability to bind to each other was analyzed using two independent HDOCK server platforms (http://hdock.phys.hust.edu.cn/, accessed on 1 September 2021) and PatchDock (https://bioinfo3d.cs.tau.ac.il/PatchDock/, accessed on 1 September 2021).

### 2.15. RNase-A-Susceptible miRNA Target Prediction and Functional Analysis

The target genes of RNase-A-susceptible miRNAs were predicted using four independent miRNA databases, namely miRmap (https://mirmap.ezlab.org/app/, accessed on 26 October 2021), miRSystem (http://mirsystem.cgm.ntu.edu.tw/, accessed on 26 October 2021), miRDB (http://mirdb.org/, accessed on 26 October 2021) and miRWalk (http://mirwalk.umm.uni-heidelberg.de/, accessed on 26 October 2021), followed by their Venn diagram analysis using the Venny 2.1 tool (http://bioinfogp.cnb.csic.es/tools/venny/index.html, accessed on 26 October 2021). The regulome of RNase-A-susceptible miRNAs and their target genes common for all used miRNA databases were reconstructed and visualized by Cytoscape 3.9.0. Functional annotation of revealed miRNA target genes was carried out by ClueGO plugin [42] using Gene Ontology (biological processes), Kyoto Encyclopedia of Genes and Genomes (KEGG), REACTOME and Wikipathways databases. Term enrichment was tested with a two-sided hypergeometric test that was corrected by the Bonferroni method. Only terms with *p* ≤ 0.05 were included in the analysis. Functional grouping and linking of the enriched terms were performed with kappa statistics (kappa score = 0.4). The regulatory network containing genes associated with biological processes related to cell adhesion was reconstructed and visualized by STRING database and Cytoscape 3.9.0, respectively.

### 2.16. Statistics

Data were statistically processed using Student’s *t*-test (two tailed, unpaired) or one-way ANOVA. Post hoc testing was completed using a post hoc Tukey test; *p* < 0.05 was considered to be statistically significant. The statistical package STATISTICA version 12.0 (Dell Technologies, Round Rock, TX, USA) was used for analysis.

## 3. Results

### 3.1. Intracellular Accumulation and Localization of RNase A–Biotin Conjugate and Its Interaction with RI in B16 and HeLa Cells

In the first step, we investigated the penetration and intracellular accumulation of RNase A into two tumor cell lines of different histogenesis, origin and malignancy. Mouse melanoma B16 is a highly aggressive melanocytic tumor cell line. Human epidermoid cervical adenocarcinoma HeLa is a cell line of epithelial origin characterized by moderate aggressiveness.

For this purpose, RNase A was conjugated with EZ-Link Sulfo-NHS-LC-Biotin, and the resulting RNase A–biotin conjugate was shown to retain the ribonuclease activity of the unaffected enzyme (see Materials and Methods, Section 2.5). B16 and HeLa cells were incubated in the presence of the RNase A–biotin conjugate for 4 h, stained with streptavidin-APC and analyzed using confocal microscopy to visualize the location of RNase A (R panel, corresponding to RNase A). Additionally, cells were stained with antibodies to RI (G panel) and NucBlue for nucleus staining (B panel) and analyzed by confocal microscopy (Figure 1a–d). Obtained data show that RNase A efficiently accumulates in the cytosol of both B16 and HeLa cells (Figure 1a,c, R panels). In B16 cells, RNase A is uniformly distributed throughout the cytoplasm, whereas in HeLa cells, a part of RNase A is localized near the nucleus (Figure 1a,c). RI is localized throughout the cytoplasm of the cells (Figure 1b,d, G panels). After penetration into B16 cells, RNase A is almost completely bound with RI (Figure 1b, Merge and Z-stack panels). Interestingly, in HeLa cells, only part of RNase A is bound by RI in the region close to the nucleus (Figure 1d, Merge panel), while near the cytoplasmic membrane, RNase A remains in a free state (Figure 1d, Z-stack panel, red signal).

The accumulation of RNase A–biotin conjugate in B16 and HeLa cells was measured using flow cytometry (Figure 1e–h). The gating strategy is shown in Appendix A. It is clearly seen that RNase A–biotin conjugate accumulated in B16 and HeLa cells with different efficiencies (compare Figure 1e,g and Figure 1f,h). The level of fluorescence intensity in B16 cells was low at the concentration of the conjugate RNase A–biotin 1 and 5 μM, and we observed no statistically reliable differences between these samples and control not only in terms of fluorescence intensity but also in terms of number of fluorescence cells. The same can be said for HeLa cells at the same concentrations of the conjugate. At low concentrations of RNase A–biotin conjugate (1–5 µM), fluorescence was detected in 3–13% of B16 cells and in 2–4% of HeLa cells (Figure 1e,f). The increase in the concentration of RNase A–biotin conjugate to 10 and 20 µM led to increases in the efficiency of its accumulation in B16 (42% and 68%, respectively) and in HeLa cells (10% and 17% of cells, respectively) (Figure 1e,f). Similarly, the levels of fluorescence intensity increase with the increase in the concentration of RNase A–biotin conjugate in the medium, reaching 13.2 and 30.0 RFU for B16 and HeLa cells, respectively (Figure 1g,h, Appendix A).

Thus, we clearly show that RNase A penetrates both B16 and HeLa cells, and the efficiency of its intracellular accumulation correlates with the concentration of RNase A in the cell medium. In B16 cells, RNase A turns out to be mostly bound with RI, whereas in HeLa cells, a noticeable amount of RNase A remains in a free state.

### 3.2. Interaction of RNase A with Ku70/Ku80 Heterodimer: Molecular Modeling and Intracellular Behavior

The Ku70/Ku80 heterodimer plays a role in DNA double-strand break recognition and repair and is expressed on the surface of different types of cells as well as localized within the nucleus and the cytoplasm. Participation of membrane-associated Ku70/Ku80 in cell–cell interaction has been reported [43,44,45]. Several studies have revealed the cytoplasmic and the cell surface localization of Ku proteins in a variety of tumor cells, including leukemia, multiple myeloma and solid tumor cell lines [44,45,46]. The data exist that Ku protein function at the cell surface is likely to be important for tumor invasion [47].

Considering all the above, we analyzed the possible role of Ku70/Ku80 heterodimer in the interaction of RNase A with tumor cells as well as its intracellular accumulation and localization. The ability of Ku70/Ku80 heterodimer to bind with RNase A was analyzed by molecular docking. For this, the crystal structures of RNase A (1FS3, RSCB Protein Data Bank) and Ku70/Ku80 heterodimer (1JEQ, RSCB Protein Data Bank) were freed from water molecules using BIOVIA Discovery Studio, and their ability to bind to each other was analyzed using two independent HDOCK server platforms and PatchDock based on different algorithms, namely the hybrid docking algorithm and the principle of surface complementarity, respectively. At the end of molecular modeling, the top 10 protein structures were visualized using Chimera/BIOVIA (Figure 2a,b).

As can be seen from the docking complexes shown in Figure 2a,b, RNase A has several binding sites on the Ku70/Ku80 heterodimer. Among these complexes, the lowest values of the binding energy ΔG and dissociation constants Kd were found for complexes in which RNase A was bound in the DNA-binding cavity of the Ku70/Ku80 heterodimer, shown in Figure 2a by a blue arrow. At the same time, a direct correlation was revealed between the depth of RNase A penetration into this structure and the values of ΔG and Kd for such complexes; this pattern was typical for both used bioinformatics platforms (Figure 2a,b, structures marked by purple frames). Thus, the performed molecular docking confirms the potential ability of RNase A to form a trimeric protein complex with the Ku70/Ku80 heterodimer that may underlie the mechanism of RNase A penetration into cells and its intracellular traffic.

Furthermore, the interaction of RNase A with Ku70/Ku80 heterodimer upon the initial stage of RNase A penetration into B16 and HeLa cells was studied by confocal microscopy (Figure 2c,d). It is seen that Ku70/Ku80 is localized throughout the cytoplasm of B16 cells and in the prenuclear space (Figure 2c, G panel). RNase A penetrates B16 cells and accumulates in the cytosol (Figure 2c, R panel), but complexes between Ku70/Ku80 and RNase A in B16 cells were not detected (Figure 2c, Merge and Z-stack panels), probably due to the low RNase A fluorescence signal.

In HeLa cells, Ku70/Ku80 localized mostly inside the nucleus and at a much lower amount in cytoplasm (Figure 2d, G panel). It was found that in HeLa cells, RNase A accumulates much more efficiently and is localized in the nucleus and cytoplasm (Figure 2d, R panel). Significant reductions in red and green signals corresponding to RNase A and Ku70/Ku80 heterodimer, respectively, were observed in the merge panel that is evidence of complex formation between Ku70/Ku80 and RNase A (Figure 2d, Merge and Z-stack panels).

Since Ku70/Ku80 can travel between the nucleus and the cytoplasm, and a large amount of this protein is found in the nucleus of HeLa cells, it is possible that RNase A also appears in the nucleus due to its translocation within the complex with Ku70/Ku80. This also cannot be excluded for B16 cells.

### 3.3. The Effect of RNase A on Intracellular RNAs

From the data obtained, it is obvious that part of RNase A in the cells remains unbound to RI, which implies that it can affect intracellular RNAs. Therefore, in the next step, we investigated by confocal microscopy the changes in intracellular RNA content in B16 and HeLa cells after incubation in the presence of RNase A and staining with SYTO RNASelect (Figure 3a,b). In intact B16 and HeLa cells, RNA was visualized inside the nucleus and throughout the cytoplasm, and in HeLa cells RNA is also clearly visualized at the border of the nucleus (Figure 3b). In the nuclei of both B16 and HeLa cells, nucleoli are clearly seen. Moreover, in HeLa cells, nucleoli are presented in a much larger amount and with a more intense fluorescence than in B16 cells.

Quantitative assessment of changes in fluorescence intensity of the cells showed that incubation of B16 and HeLa cells with RNase A at concentrations of 1 and 5 μM for 4 h already reduced total RNA content by 20–30% for B16 and by 10–35% for HeLa cells (Figure 3c,d). The increase in RNase A concentration to 20 μM caused reductions in total RNA content at the 4 h time point by 40% and 25% for B16 and HeLa cells, respectively. Prolonged incubation of the cells with RNase A (48 h) resulted in the significant decrease in the fluorescence intensity (Figure 3c,d): at RNase A concentrations of 1 and 5 μM, total RNA content in B16 cells decreased by 70% and reached 80% at the RNase A concentration of 20 μM. In HeLa cells at the 48 h time point, RNA content essentially decreased by 55% even at the lowest concentration of RNase A used (1 μM). An interesting fact is that RNase A decreasing the total RNA content in the cells also decreased the number of nucleoli in the nucleus and their fluorescence intensity, as is clearly seen for HeLa cells (Figure 3b).

Thus, we clearly show that RNase A being in the cells decreases their RNA content. As can be seen from the presented data, RNase A reduces the RNA content to a greater extent in B16 cells than in HeLa cells. This is due to several reasons. First, RNase A penetrates better into B16 than HeLa cells (compare Figure 1c,e and Figure 1d,f). Moreover, after penetration into B16 cells, RNase A is localized in the cytosol, and after penetration into HeLa cells, it is localized to a greater extent in the nucleus. Thus, in B16 cells, RNase A mainly decreases the content of cytosolic RNA, whereas in HeLa cells it reduces the content of rRNA expressed in nucleoli and decreases the number/fluorescence intensity of nucleoli.

### 3.4. The Effect of RNase A on the Migration Activity, Adhesion, Invasion and Colony Formation of B16 and HeLa Cells

The most important parameters that characterize the invasive potential of tumor cells include migration, motility, invasion, adhesion and colony formation abilities. The influence of RNase A on the migratory activity of B16 and HeLa cells was investigated in vitro via scratch assay. The integrity of the cell monolayer was disrupted by scratching, and the scratch area was measured in the absence of the enzyme (control) and in the presence of different concentrations of RNase A. The migration width was estimated as the difference between scratch width filled with cells after 48 h and the initial scratch width (Figure 4a–d).

Incubation of B16 cells in the presence of RNase A caused a decrease in the cell migration in a time-dependent manner: at the 24 h time point, scratch areas were 60% in experimental wells and 18% in control wells (Figure 4a,b). At the 48 h time point, the B16 migration was still decreased relative to the control: 20–25% and 5% for experimental and control wells, respectively. No effect of RNase A on B16 cell migration at 72 h was observed regardless of RNase A concentration (Figure 4a,b). Interestingly, incubation of HeLa cells with RNase A had almost no influence on the migration of these cells at the 24 h and 48 h time points. Only at 72 h, a slight dose-dependent decrease in the migration activity of cells was observed: the area of free scratch was 75% for HeLa cells treated with RNase A and 55% for the control cells (Figure 4c,d).

The motility of tumor cells plays an important role in metastasis formation. We hypothesized that RNase A can affect this process in B16 and HeLa cells. The cell motility of B16 and HeLa cells in the presence of nontoxic concentrations of RNase A (2.5, 5 and 10 µg/mL) was measured by FBS- and chemoattractant-stimulated trans-well cell penetration assay for 48 h under standard conditions using the xCELLigence system. As depicted in Figure 4e,f, motility of B16 cells did not change in the presence of RNase A, whereas the migration activity of HeLa cells was 1.3-fold suppressed.

The effect of RNase A on the invasion potential of B16 and HeLa cells was monitored in a 16-well CIM plate precoated with Matrigel in real-time mode. We found that the invasion of B16 and HeLa cells was similarly slowed down (by a factor of two) by RNase A (Figure 4g,h). Surprisingly, the slowly scratch-filling HeLa cells were highly motile and invasive compared to B16 cells in trans-well tests. Observed inconsistencies can be explained by differences in experiment conditions: in the case of scratch tests, we observed basal 2D migration of the cells in the absence of any stimulators, whereas in the case of trans-well 3D tests, migration and invasion were enhanced by chemoattractant (10% FBS). It was found that RNase A at the concentrations of 5 and 10 μg/mL resulted in statistically significant enhanced cell adhesion of both B16 and HeLa cells by 20% compared to the control group (Appendix A).

Clonogenic activity of B16 and HeLa cells, which determines the capacity of single cells to grow into a colony and is associated with the stemness of tumor cells [48], was compared with cells exposed to RNase A used at different concentrations. B16 and HeLa cells were seeded at a low density (200 cells/well) in DMEM supplemented with 10% FBS in the presence of RNase A and incubated for 8 and 14 days, respectively, followed by colony visualization. It turned out that RNase A at the concentrations of 5 and 10 μg/mL only slightly reduced the clonogenicity of both B16 and HeLa cells, and this effect did not exceed 12% (Appendix A).

### 3.5. The Influence of RNase A on Metastasis of Melanoma B16 In Vivo

The ability of RNase A to affect metastasis was studied using a metastatic melanoma B16 model without a primary tumor node. The scheme of the experimental setup is depicted in Figure 5a. The B16 metastatic model was induced in mice by i.v. injection of B16 cells into the lateral tail vein of C57Bl mice (Figure 5a). Starting from day 4 after tumor implantation, the animals were assigned to three groups: 1 (C) —control injected i.m. with saline buffer; 2 and 3—injected i.m. with RNase A at the doses of 0.7 and 7 μg/kg, respectively, according to the scheme 5 + 2 for two weeks.

The doses used in the study were chosen according to our previous study, which demonstrated efficient inhibition of metastasis [20,23]. During the experiment, the weight of the mice was measured; however, no significant differences between the groups were found, indicating the absence of systemic toxic effects of RNase A in the doses used (Appendix A). One hour after the last injection, animals were sacrificed, and blood samples, lung tissues from healthy C57Bl/6 mice, metastatic foci and adjacent lung tissue from experimental groups were collected for subsequent histological analysis and the evaluation of miRNA expression levels.

RNase A efficiently inhibited metastatic spreading of melanoma B16. As shown in Figure 5b,c, the administration of RNase A resulted in a significant decrease in the number of surface and internal metastases in the lungs of mice with B16 compared with the control animals. In the control group, the average number of surface metastatic foci was 60.1 ± 8.6, while in experimental groups treated with RNase A at the doses of 0.7 and 7 μg/kg (here and after 0.7R and 7R groups, respectively) we observed sevenfold and 3.2-fold decreases in the number of surface metastases amounting to 8.6 ± 1.7 and 18.6 ± 4.3, respectively (Figure 5b).

Although metastatic foci in the lungs of control and experimental mice demonstrated approximately the same structure—they had rounded shape with clear boundaries, were located predominantly around the blood vessels and bronchi and were represented by the polymorphic or spindle-shaped atypical cells containing the brown pigment melanin—the areas occupied by metastases in the experimental group were much smaller (Figure 5d). RNase A administration reduced the average metastases area 3.7-fold and 2.2-fold based on the counting of the area occupied by internal metastases, and MIIs (metastatic inhibition index) were 73.1 ± 9.2 and 54.4 ± 12.3% for groups 0.7R and 7R, respectively (Figure 5c).

Thus, for the first time, we demonstrated the ability of RNase A to suppress the metastases development of melanoma, which is characterized by high aggressiveness and dissemination rate.

### 3.6. The Effect of RNase A on the Expression of EMT Markers in Metastatic Foci and Adjacent Lung Tissue of B16 Melanoma-Bearing Mice

In order to evaluate the effect of RNase A on metastasis development, we investigated the expression of markers associated with epithelial–mesenchymal transition (EMT): Cdh1 (E-cadherin), Tjp1, Fn and Vim (vimentin), which play important roles in cancer invasion and metastasis. The expression of E-cadherin and vimentin was investigated using fluorescence-based immunohistochemistry. As depicted in Figure 6, RNase A administration enhanced the expression of E-cadherin in lung metastasis foci: the fluorescence intensity was increased 2–2.3-fold in experimental groups 0.7R and 7R compared to the control (Figure 6a). Moreover, simultaneously with the increase in E-cadherin expression, we observed a 2.5–2.6-fold decrease in the expression of vimentin in metastatic foci of the same groups in comparison with the control animals (Figure 6a). It is worth mentioning that the effects of RNase A on EMT markers’ expression in metastatic foci at the dose of 0.7 μg/kg were more pronounced than at the dose of 7 μg/kg.

It should be noted that in metastatic foci, the level of Cdh1 expression was 0.5, whereas the expression levels of other EMT markers were about one (Figure 6b). In the groups 0.7R and 7R, a 1.6-fold increase in the expression level of Cdh1 and no effect on Tjp1 or Fn expression were detected. At the same time, Vim expression was slightly decreased (Figure 6b). An increase in the RNase A dose to 7 μg/kg resulted in a 2.4-fold increase in Cdh1 expression and a 1.4-fold decrease in Tjp1 expression in metastatic foci; moreover, the Vim level was still down, and the Fn level was unaffected (Figure 6b).

In addition to metastatic foci, we were able to evaluate the level of EMT markers’ expression in the lung tissue adjacent to metastatic foci and compare with the level in the lung tissue of healthy animals, because they have the same origin (epithelial) in contrast to metastatic foci having a neuroectodermal origin. Interestingly, the lung tissue adjacent to metastatic foci was characterized by a significant (threefold) decrease in the level of Cdh1 relative to the healthy control, 1.3-fold increases in the expression levels of Tjp1 and Fn and a 1.5-fold decrease in the Vim expression level (Figure 6c). In the group 0.7R (RNase A dose 0.7 μg/kg), a strong upregulation of Cdh1 expression to the level of healthy animals was observed, Vim expression was slightly increased and Tjp1 and Fn levels were not changed (Figure 6c). An increase in the RNase A dose to 7 μg/kg led to some decreases in Cdh1, as well as decreases in Tjp1 and Fn expression levels (but statistically insignificant compared to the 0.7R group, which nevertheless reached the level of healthy animals (Figure 6c). As in the 0.7R group, Vim expression in the 7R group was slightly increased, reaching the level of healthy animals (Figure 6c).

### 3.7. Alteration of miRNA Expression in B16 and HeLa Cells In Vitro and in B16 Metastatic Foci and Adjacent Lung Tissues In Vivo under the Action of RNase A

Recently, we have shown that miRNA circulating in the blood stream and found in tumor tissue represents one of the main classes of molecular targets of RNase A in vivo [23]. To answer the question of what the effects of RNase A are on miRNA expression patterns in vitro, we first analyzed changes in miRNA expression in B16 and HeLa cells after incubation with the enzyme. The miRNA panel to be analyzed included miR-21a, miR-10b, miR-145a, miR-31, let-7g and miR-155 (Table 1). These miRNAs are in the list of the top 50 miRNAs changing in blood serum and tumor tissue of mice with Lewis lung carcinoma under the action of RNase A, according to NGS data [23]. The alterations in the expression of these miRNAs under the action of RNase A were observed in the case of lymphosarcoma RLS_40_ in CBA mice [49]_._ The sequences of these miRNAs were checked using MirBase and were found to be identical in mice and humans (Appendix A).

All chosen miRNAs were noticeably expressed in B16 and HeLa cells (Table 1). Five miRNAs in B16 cells are characterized by expression levels of 0.7–0.8, except miR-155, which was expressed at the level of 0.56 (Table 1). In HeLa cells, similar levels of expression of 0.78–1 were detected for all six miRNAs (Table 1). Treatment of B16 cells with RNase A (5 µg/mL) led to a 1.2–1.7-fold decrease in the levels of miR-10b and miR-155 compared to the control (Table 1 and Figure 7a), while the levels of let-7g, miR-21a, miR-31 and miR-145a did not change (Figure 7a). In HeLa cells, levels of tested miRNA remained unaltered after RNase A treatment (Table 1 and Figure 7b).

Entirely different miRNA expression patterns were observed in vivo (Figure 7c–e). In metastatic foci, RNase A administration caused a decrease in the expression levels of all six miRNAs, but the differences in the expression levels were less pronounced due to the initially rather low levels of these miRNAs in the metastatic foci, excluding let-7g (Figure 7c). The expression level of let-7g in metastatic foci was rather high, and RNase A administration led to a fourfold decrease in the let-7 level (Figure 7c).

In the lung tissue of healthy animals, which was used as a control for tissue adjacent to metastatic foci, six miRNAs selected for analysis were detected at levels of 0.2–1 (Figure 7d). Interestingly, the miRNA profile in healthy tissue and tissue adjacent to metastatic foci significantly differs. Increases in the levels of five miRNAs in the adjacent tissue in comparison with the healthy group were found: miR-21a, 1.5-fold; miR-31, 2.4-fold; miR-155, fourfold; miR-145a, 2.8-fold; miR-10b, 4.5-fold (Figure 7d). The level of let-7g remained similar to the level in healthy lung tissue (Figure 7d). In experimental groups treated with RNase A at the doses of 0.7 and 7 μg/kg (0.7R and 7R groups), the expression levels of miR-21a, miR-31, miR-145a and miR-155 were decreased compared to the control group (without any treatment) (Figure 7d). In the 7R group, the levels of miR-21a, miR-31 and miR-145a dropped to the levels of the healthy control, and the level of let-7g even dropped below the level of the healthy group (Figure 7d).

In the blood serum of healthy animals, six miRNAs selected for analysis were detected at levels of 0.2–0.4 (Figure 7e). Progression of melanoma resulted in pronounced increases in the levels of all miRNAs. Administration of RNase A at a dose of 0.7 µg/kg caused a significant decrease in the level of miR-21a by 3.5-fold, miR-10b by 4.5-fold and miR145a and let-7g by 1.3-fold (Figure 7e). Increasing the dose of RNase A to 7 μg/kg led to a decrease in the level of miR-21a and miR-10b by 11-fold and miR-145a and let-7g by 2.1–2.3-fold, while the levels of miR-31 and miR-155 remained unchanged (Figure 7e).

### 3.8. RNase-A-Susceptible miRNA: Target Prediction and Functional Analysis

Given the revealed ability of RNase A to significantly affect the expression of miRNAs, including miR-21a, miR-31, miR-145a, miR-155 and miR-10b, in B16 melanoma-bearing mice (Figure 7), we next questioned the expression of which genes can be regulated by these RNase-A-susceptible miRNAs and in what biological processes these genes can be involved. To understand this, target genes of mentioned miRNAs were predicted using four independent miRNA databases: miRmap, miRSystem, miRDB and miRWalk. Then, using the Venn diagram approach, overlapping genes for each analyzed miRNA common for miRNA databases were identified (Figure 8a, white regions).

Reconstruction of the regulome, containing RNase-A-susceptible miRNAs and their revealed target genes, showed its clustered architecture: all miRNAs were associated predominantly with their own list of target genes, and only 29 genes were identified as common for several miRNAs (Figure 8b). Interestingly, oncosuppressor miR-145a was found to be the most enriched by target genes, including 20 common regulators, compared to oncomirs miR-21a, miR-10b, miR-31 and miR-155. Next, in order to understand the intracellular processes in the regulation of which the revealed target genes can be involved, their functional analysis was further carried out. The obtained results demonstrated that RNase-A-susceptible miRNAs can modulate a wide range of processes and signaling pathways associated with tumor cell proliferation and motility (Figure 8c). It was found that the majority of identified terms were related to the processes that had already been revealed as sensitive to RNase A treatment, such as regulation of cell adhesion, morphogenesis of epithelium and organization of lamellipodium and intercellular connections.

## 4. Discussion

In this work, to study the antitumor potential of RNase A in detail, we focused on four main issues: (1) the accumulation of RNase A and its intracellular localization, including co-localization with RI in B16 and HeLa cells, the role of Ku70/Ku80 heterodimer in intracellular accumulation and traffic of RNase A and the ability of RNase A to degrade intracellular RNA; (2) the ability of RNase A to affect invasion properties of B16 and HeLa cells in vitro and B16 melanoma in vivo, including evaluation of EMT markers in metastatic foci; (3) a search for correlations between the antitumor effects of RNase A in vitro and in vivo and changes in the miRNA profile in tumor cells, blood serum metastatic foci and adjacent tissues; and (4) building a regulatory network that can explain the observed antitumor effects.

We demonstrated that RNase A efficiently accumulated in B16 and HeLa cells having neuroectodermal and epithelial origins, respectively. Previously, it has been shown that the selectivity of RNase A towards tumor cells results from its interaction with abundant cell surface proteoglycans containing glycosaminoglycans, such as heparan sulfate and chondroitin sulfate, as well as with sialic-acid-containing glycoproteins [50]. It was demonstrated using HeLa cells that the cellular entry of RNase A is realized through macropinocytosis and clathrin-mediated endocytosis similar to that of the cell-penetrating peptides [51]. On the other hand, RNase A was found to not accumulate in monolayer culture of human colorectal cancer HT-29 cells [52].

The absence of cytotoxic activity of RNase A is associated with its tight binding inside the cell by RI [17,53,54]. RI binds multiple members of the RNase A superfamily with equilibrium dissociation constants in the femtomolar range, forming one of the tightest non-covalent complexes among biomolecules [55]. Nevertheless, the closest relative of RNase A, angiogenin, which is able upon entering the cells to be bound to the cytosolic RI, is undergoing phosphorylation by protein kinase C and cyclin-dependent kinase, enabling angiogenin to evade RI and enter the nucleus [56]. Our data demonstrated that in HeLa cells, a noticeable amount of RNase A remains unbound with RI. In B16 cells, a lesser part of the RNase A compared to HeLa cells also remains unbound. The fact that RNase A remains unbound by RI and catalytically active after intracellular accumulation is confirmed by effective RNA degradation inside both B16 and HeLa cells; this effect in the nucleus of HeLa cells is more pronounced (Figure 3).

It should be mentioned that a reduction in the amount of intracellular RNA in tumor cells caused by RNase A did not lead to cell death. The choice between life and death of a cell subjected to exogenous RNases depends on the characteristic pattern of hydrolysis products of the cellular RNA, which reflects the results of complicated interactions between molecular determinants of RNases on one side [6,57] and the cellular targets on the other [58,59,60]. It is more likely that after the cleavage of intracellular RNAs by RNase A, a new pool of RNAs (including regulatory RNAs) and novel RNA regulation networks are formed.

Ku70/Ku80 heterodimer has been found to associate with RNA helicase A (RHA) [61], a multifunctional protein, which is involved in several steps of RNA metabolism, such as RNA processing, cellular transit of viral molecules, ribosome assembly, regulation of transcription and translation of specific mRNAs [62]. We obtained interesting data on the role of the Ku70/Ku80 heterodimer in the penetration and intracellular traffic of RNase A. The performed molecular docking confirms the ability of RNase A to form a trimeric protein complex with Ku70/Ku80 heterodimer, and this may indicate that RNase A penetrates cells and is translocated to the nucleus using Ku70/Ku80 as a vehicle. The fact that Ku70/Ku80 is localized throughout the cytoplasm of both tumor cell lines and in the nuclei of HeLa cells, as along with the fact of its co-localization with RNase A, confirms this (Figure 2). Moreover, the observed decrease in the number and intensity of fluorescent signals of nucleoli in the RNase-A-treated cells is evidence of rRNA degradation by RNase A after its translocation to nuclei.

We demonstrated in this work that RNase A affects the invasiveness of tumor cells in vitro and in vivo. RNase A significantly reduced migration, motility and invasion of tumor cells in vitro, while the effects of the enzyme on adhesion and colony formation were rather small. In vivo RNase A efficiently suppressed metastases development in the lungs of mice with metastatic melanoma, which was accompanied by an alteration of the expression of some EMT markers in metastatic foci and adjacent tissue.

Epithelial–mesenchymal transition (EMT) involves modifications of gene expression, leading to the disruption of the epithelial phenotype and the building of the mesenchymal phenotype. EMT is an important event that regulates tumor invasion and metastasis [63,64]. Cadherin and vimentin are important genes of the EMT process. E-cadherin is highly expressed in epithelial cells and represents the extensive array of tight junctions closely holding the cancer cells to each other [65]. Reduced E-cadherin functionality through mutation, degradation, etc., is an essential contributor to metastasis progression in multiple different epithelial-based malignancies [66]. Vimentin, a major intermediate silk filament protein present in stromal cells, participates in cancer cells’ adhesion, invasion, migration and signaling by regulating the interaction between cytoskeletal proteins and cell adhesion molecules [67]. In this study, it was shown that RNase A administration led to an increase in the level of Cdh1 and decreases in the levels of Tjp1 and Vim in metastatic foci, which indicates a decrease in the invasive potential of metastatic cells. A similar pattern was observed for these markers in the lung tissue adjacent to metastatic foci, and the decrease in Fn expression was also observed. As a result, the adjacent tissue according to the profile of expression of EMT markers acquires a phenotype similar to that of healthy tissue, which indicates that RNase A disrupted the formation of a favorable tumor microenvironment.

The study of alterations of miRNA expression in vitro in the cells treated with RNase A revealed the decrease in miR-10b and miR-155 expression in B16 cells and no effect of RNase A on the tested miRNAs in HeLa. At the same time, the in vivo picture was significantly different. As expected, dose-dependent decreases in the levels of most miRNAs except for miR-31 and miR-155 in the blood serum of mice with B16 after RNase A administration were observed.

One of our main findings is that we were able to identify the differences in miRNA expression profiles between metastatic foci and adjacent tissues as well as between adjacent conditionally normal tissue and lung tissues collected from healthy animals. Interestingly, in adjacent conditionally normal tissues, we observed much higher levels of miRNAs in comparison with the healthy group and decreases in these miRNA levels after RNase A administration in a dose-dependent manner. It should be noted that in metastatic foci of the control group, miRNA levels were 2–3-fold lower in comparison with adjacent tissue. In metastatic foci, RNase A had no effect on the expression of analyzed miRNAs; their expression remained approximately at the same level. The exception was let-7g, whose level significantly decreased after the treatment with RNase A.

Our results demonstrate that the antitumor potential of RNase A is tightly associated with the suppression of EMT-related processes. RNase A was found to effectively inhibit motility (Figure 4) and enhance the adhesiveness (Appendix A) of tumor cells as well as block their metastasis (Figure 5); moreover, RNase A significantly affects the expression of key EMT markers not only in B16 melanoma metastatic foci but also in the tissue surrounding them (Figure 6). Considering the high enrichment of a range of EMT-related terms in the pathway network of RNase-A-susceptible miRNA target genes (Figure 8c), we next questioned whether these miRNAs could be involved in the regulation of anti-EMT activity of RNase A. To understand this, a regulatory network consisting of (i) RNase-A-sensitive miRNAs (Figure 7), (ii) their target genes included in EMT-related functional terms (Figure 8c) and (iii) RNase-A-susceptible EMT-associated markers (Cdh1, Fn1, Tjp1, Vim) was reconstructed (Figure 9). The obtained data indeed demonstrate that the inhibitory effect of RNase A on a panel of miRNAs in metastatic foci and adjacent tissue can underlie observed changes in the expression of EMT-related key genes in these compartments: the level of analyzed EMT markers can be regulated directly by RNase-A-susceptible miRNAs (miR-10b, miR-31, miR-145a) or indirectly by a range of miRNA-targeted genes predicted by our in silico approach (Arf6, Ctnnd1, Fzd7, Gdnf, Smad3, Pxn, Ret) (Figure 9). Since these findings were based on computational analysis, their further experimental verification is required.

Based on the obtained data, we hypothesized a potential mechanism of antitumor activity of RNase A (Figure 10). Obviously, this scheme presents a simplified view of the anticancer action of RNase A, consisting of binding to a tumor cell, entering the cytosol, degrading intracellular RNAs, including rRNAs and miRNAs, followed by the suppression of migration and invasion of tumor cells, activation of adhesion and control cell junction and inhibition of metastases. Taking into account that RNase A caused the degradation of miRNAs in the bloodstream, we cannot exclude the effect on these events of the changed miRNA profile in the bloodstream. The downstream mechanisms of RNase A antitumor activity may include the alteration of miRNA expression, downregulation of tumor suppressors and oncomirs, upregulation of adhesion genes, control and organization of the cell junction and finally the attenuation of tumor malignancy and metastasis.

Unraveling these mechanisms requires a detailed investigation of the expression profiles in cells treated with RNases. The most immediate challenge in the further exploration of bovine pancreatic RNase as a platform for clinical applications is to better understand its mechanism of action and to use this knowledge to improve its selectivity against malignant cells.

## Figures and Tables

**Figure 1 pharmaceutics-14-01173-f001:**
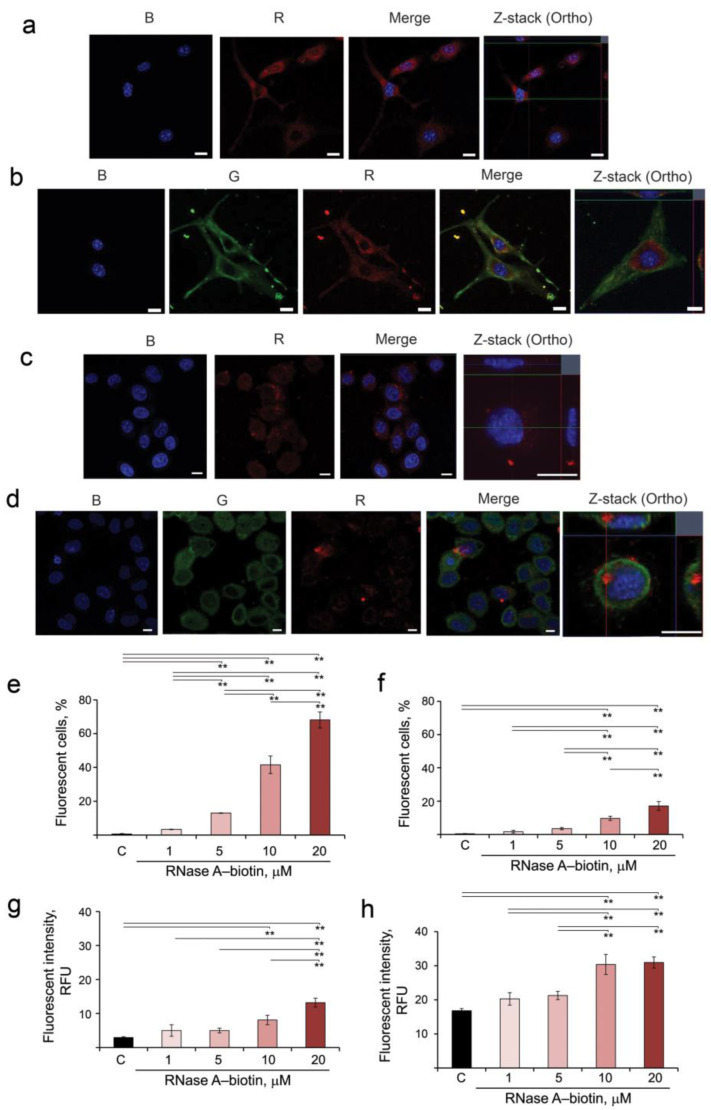
Accumulation of RNase A in B16 and HeLa cells, its intracellular localization and co-localization with RI. (**a**,**b**) B16 cells. (**c**,**d**) HeLa cells. B16 and HeLa cells were incubated in the presence of RNase A–biotin conjugate (20 µM) for 4 h followed by staining with streptavidin-APC, anti-RI rabbit antibodies and goat anti-rabbit antibodies conjugated with Alexa Fluor 488 and analyzed using confocal microscopy. Fluorescence in the blue channel (B panel) corresponds to NucBlu (nuclei staining), the green channel (G panel) corresponds to RI stained by Alexa Fluor 488 and the red channel (R panel) corresponds to RNase A–biotin conjugate stained with streptavidin-APC. Scale bars: 10 µm. (**e**,**g**) Accumulation of RNase A in B16 cells. (**f**,**h**) Accumulation of RNase A in HeLa cells. B16 and HeLa cells were incubated with the RNase A–biotin conjugate at concentrations of 1, 5, 10 and 20 µM at 37 °C for 4 h and analyzed using flow cytometry. Data were statistically analyzed using one-way ANOVA with a post hoc Tukey test and are presented as mean ± S.E.M.; ** *p* < 0.01.

**Figure 2 pharmaceutics-14-01173-f002:**
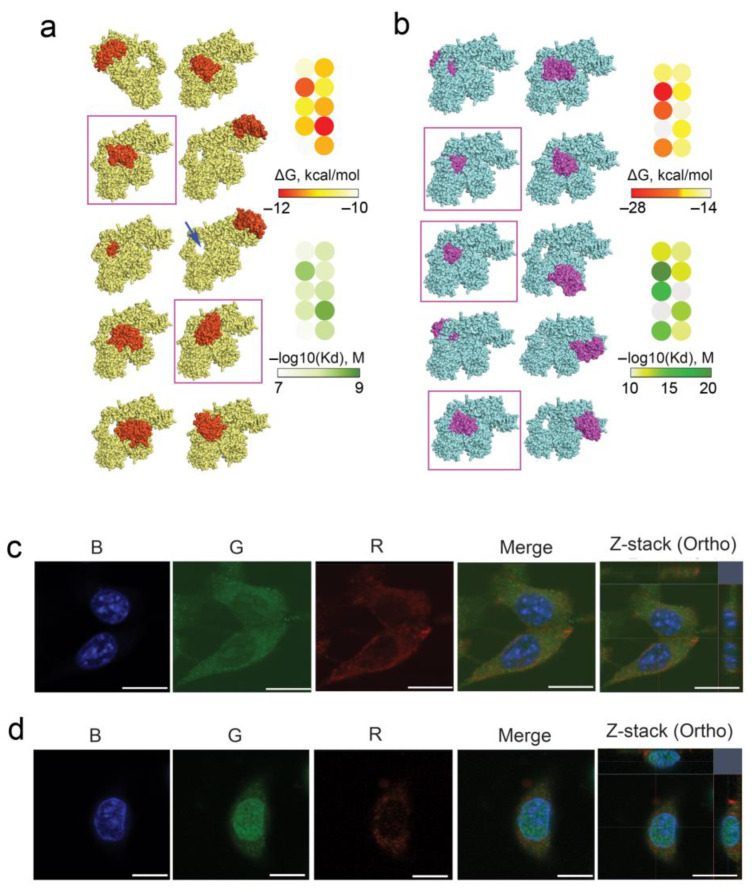
Interaction of RNase A with Ku70/Ku80 heterodimer. (**a**,**b**) Structures of the complex between RNase A and Ku70/Ku80 obtained by molecular modeling using bioinformatics platforms HDOCK server and PatchDock, respectively. The ΔG values of RNase A binding to Ku70/Ku80 heterodimer and corresponding dissociation constants (Kd) were calculated for each obtained structure using the PRODIGY resource. Heterodimer Ku70/Ku80 is yellow (**a**) or blue (**b**), and RNase A is orange (**a**) or purple (**b**). The arrow (**a**) shows the DNA-binding loop formed by Ku70/Ku80 heterodimer. Docking complexes characterized by the smallest values of ΔG and Kd are highlighted with purple frames. (**c**,**d**) Accumulation of RNase A in B16 and HeLa cells, respectively, and its co-localization with Ku70/Ku80 heterodimer. B16 and HeLa cells were incubated in the presence of RNase A–biotin conjugate (20 μM) for 4 h followed by staining with streptavidin-APC, Ku70/Ku80 monoclonal antibody and secondary antibody with Alexa Fluor Plus 488 and analyzed by confocal fluorescence microscopy. Fluorescence in the blue channel (B panel) corresponds to DAPI (staining of nuclei), the green channel (G panel) corresponds to Ku70/Ku80 Alexa Fluor Plus 488 and the red channel (R panel) corresponds to RNase A–biotin conjugate stained with streptavidin-APC. Scale bars: 10 µm.

**Figure 3 pharmaceutics-14-01173-f003:**
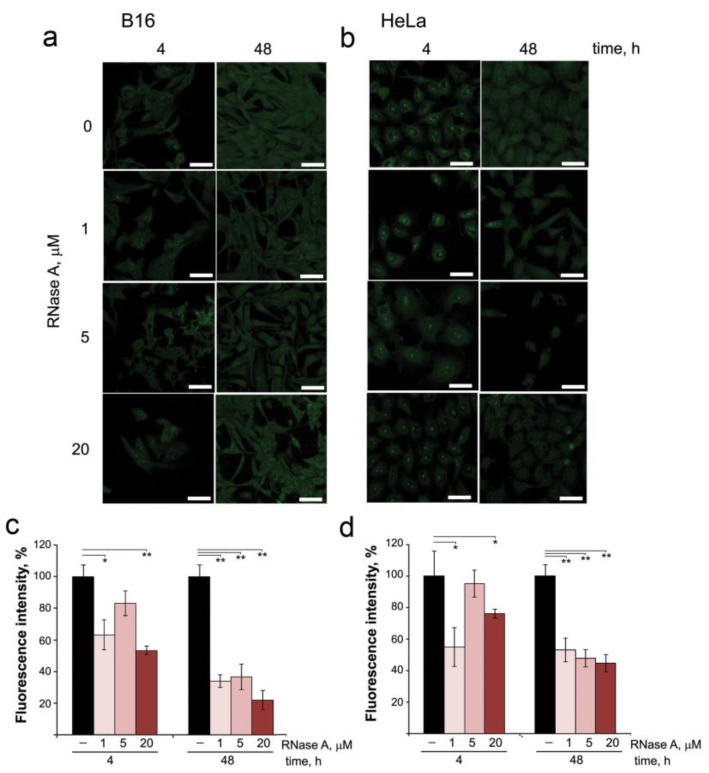
The effect of RNase A on the content of intracellular RNAs in B16 and HeLa cells. (**a,b**) Images of B16 and HeLa cells, respectively, after incubation for 4 or 48 h in the presence of RNase A (1, 5 and 20 µM). Cells were stained with SYTO RNASelect and analyzed using confocal microscopy. (**c**,**d**) The fluorescence intensity of intracellular RNAs in B16 and HeLa cells, respectively, evaluated using ImageJ software. Data are presented as mean ± S.E.M. Statistical processing of the data was carried out using Student’s *t*-test; * *p* < 0.05, ** *p* < 0.01. Scale bars: 10 µm.

**Figure 4 pharmaceutics-14-01173-f004:**
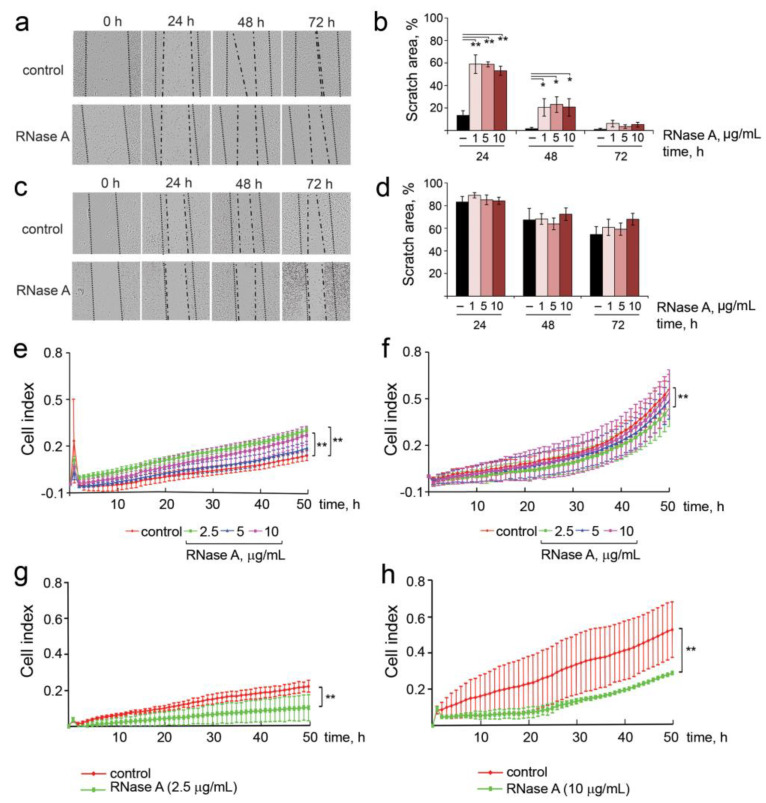
Migration and invasion of B16 and HeLa cells incubated in the presence of RNase A. (**a**,**c**) Scratch healing assay for B16 and HeLa cells, respectively. RNase A concentration was 10 µg/mL. Incubation time is shown at the top (4× magnification). Dotted line shows initial edge of the scratch and dash-and-dot line shows boundary of the monolayer 24, 48 and 72 h after scratching. (**b**,**d**) Quantification of the data of scratch healing assay for B16 and HeLa cells, respectively. (**e**,**f**) FBS-induced trans-well penetration (motility) of B16 and HeLa cells exposed to RNase A (2.5, 5 and 10 µg/mL) for 48 h, respectively. (**g**,**h**) FBS-induced trans-well Matrigel barrier penetration (invasion) of B16 and HeLa cells exposed to RNase A at concentrations of 2.5 and 10 μg/mL, respectively. Cell motility and the rate of invasion were monitored using the xCELLigence system. Data are presented as mean ± S.E.M. Data were statistically processed using Student’s *t*-test; * *p* < 0.05, ** *p* < 0.01.

**Figure 5 pharmaceutics-14-01173-f005:**
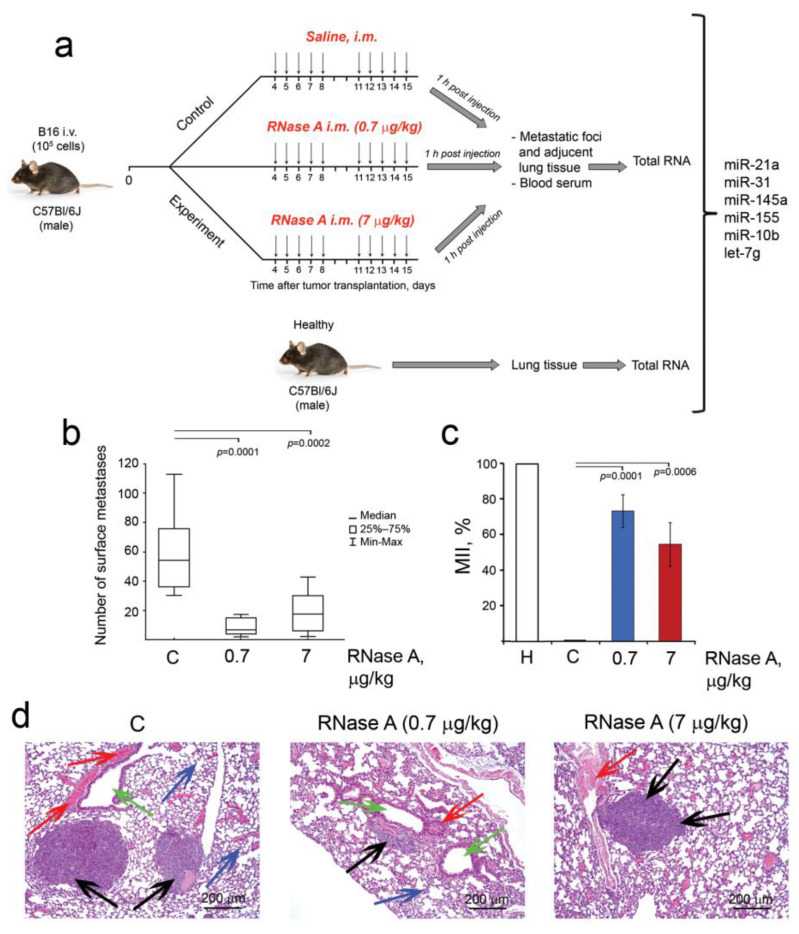
The influence of RNase A on the metastasis development in B16 melanoma-bearing mice. (**a**) Design of the experiment. B16 cells (10^5^ cells, 0.2 mL) were i.v. implanted into C57Bl/6 mice. Starting on day 5 after B16 implantation, animals received saline buffer or RNase A (0.7 and 7 µg/kg) i.m. daily, except weekends. On day 15, mice were sacrificed 1 h after the last injection, and the lungs were collected for calculation of surface and internal metastases. Metastatic foci and adjacent lung tissue, lung tissues from healthy C57Bl/6 mice and blood samples were collected; total RNA was isolated and used for miRNA analysis. (**b**) Number of surface lung metastases. (**c**) Metastasis inhibition index (MII). Data were statistically analyzed using one-way ANOVA with a post hoc Tukey test and are presented as median (**b**) and as mean ± SE (**c**). Statistical significance: *p* ≤ 0.05. (**d**) Representative histological images of the lungs of B16 melanoma-bearing mice treated with RNase A. Metastatic foci, adjacent lung tissue, blood vessels and bronchi are indicated by black, blue, red and green arrows, respectively. Hematoxylin and eosin staining, original magnification ×100. C–mice with B16 treated with saline buffer.

**Figure 6 pharmaceutics-14-01173-f006:**
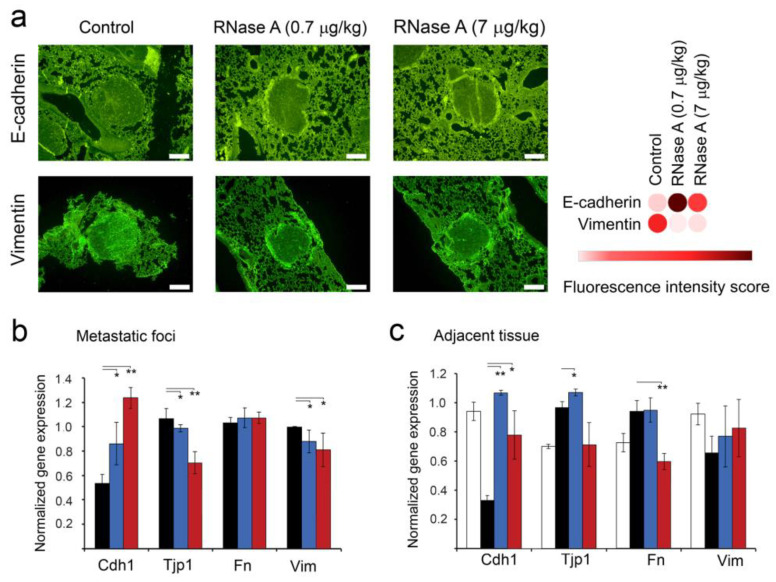
The effect of RNase A on the expression of EMT-associated markers in metastatic foci and adjacent lung tissue of mice with melanoma B16. (**a**) Representative images of fluorescence-based immunohistochemical staining of lung metastases with primary antibodies against E-cadherin and vimentin followed by incubation with secondary antibodies conjugated to Alexa Fluor^®^ 488 (magnification ×200). Green fluorescence intensities corresponding to expression of E-cadherin and vimentin were calculated for each image using ImageJ software and visualized as a heat map using the Morpheus instrument. (**b**,**c**) The levels of expression of EMT-associated genes in metastatic foci and adjacent lung tissue, respectively. Data of RT-qPCR. Gene expression levels were normalized to HPRT. Data are presented as mean ± S.E.M. Data were statistically processed using Student’s *t*-test; * *p* < 0.05, ** *p* < 0.01.

**Figure 7 pharmaceutics-14-01173-f007:**
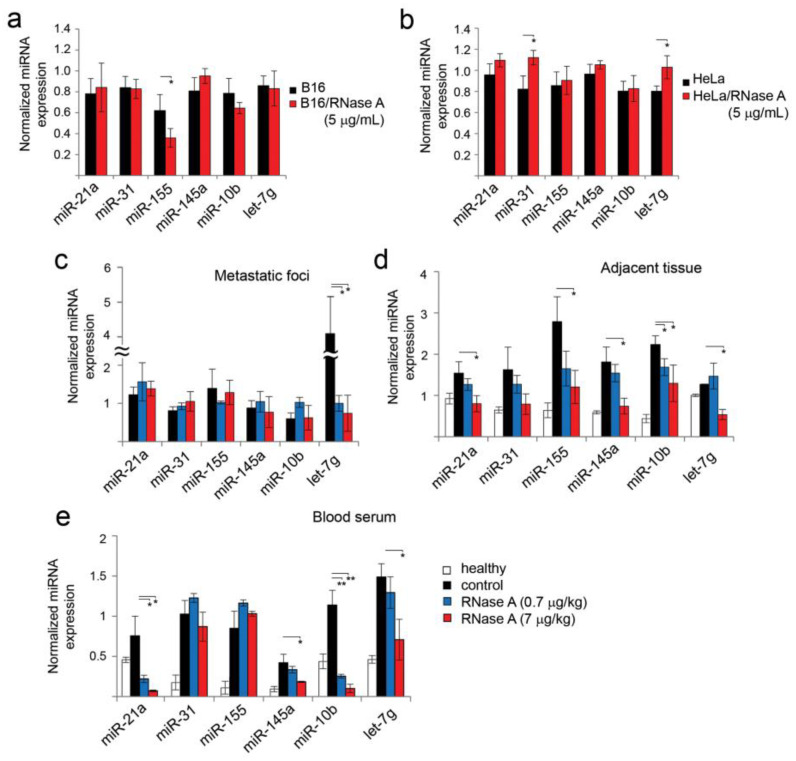
Influence of RNase A on miRNA expression in HeLa and B16 cells in vitro and in vivo. RT-qPCR data. Expression of miRNAs in B16 (**a**) and HeLa (**b**) cells incubated with RNase A (5 μg/mL) for 48 h under standard conditions. The levels of miRNA in metastatic foci (**c**), adjacent tissue (**d**) and blood serum (**e**) of mice with melanoma B16 received daily RNase A at the doses of 0.7 and 7 μg/kg. The level of miRNA expression in lung tissue and metastatic foci was normalized to U6 snRNA. Data of miRNA levels in healthy lung tissue were used as control for tissue adjacent to metastatic foci. The concentration of miRNA derived from serum was normalized to serum volume. miRNA levels in blood serum of healthy mice did not exceed 0.2–0.4. Data are presented as mean ± S.E.M. Data were statistically processed using Student’s *t*-test; * *p* < 0.05, ** *p* < 0.01.

**Figure 8 pharmaceutics-14-01173-f008:**
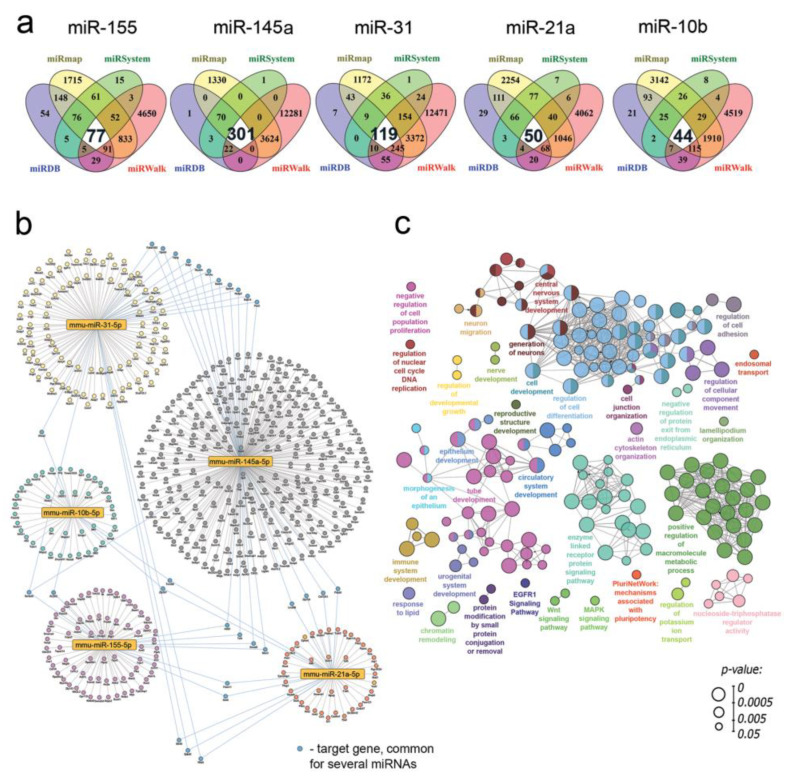
Functional annotation of RNase-A-susceptible miRNAs identified in B16 cells. (**a**) Venn diagram analysis of target genes of five RNase-A-susceptible miRNAs. For each miRNA, target genes were obtained from four independent miRNA databases including miRmap (yellow), miRSystem (green), miRDB (blue) and miRWalk (red). Number of genes common for all analyzed miRNA databases are labeled in white circles with bold numbers. (**b**) The regulome of RNase-A-susceptible miRNAs and their target genes (labeled in bold numbers in Venn diagram depicted in Figure 8a). The network was reconstructed and visualized by Cytoscape 3.9.0. (**c**) The interaction network of significant terms enriched with revealed miRNA target genes. Functional annotation was performed by ClueGO plugin by using GeneOntology (biological processes), KEGG, REACTOME and Wikipathways databases. The functionally grouped network was linked based on the kappa score of terms. Only terms/pathways with *p* < 0.05 after the Bonferroni correction were included in the network.

**Figure 9 pharmaceutics-14-01173-f009:**
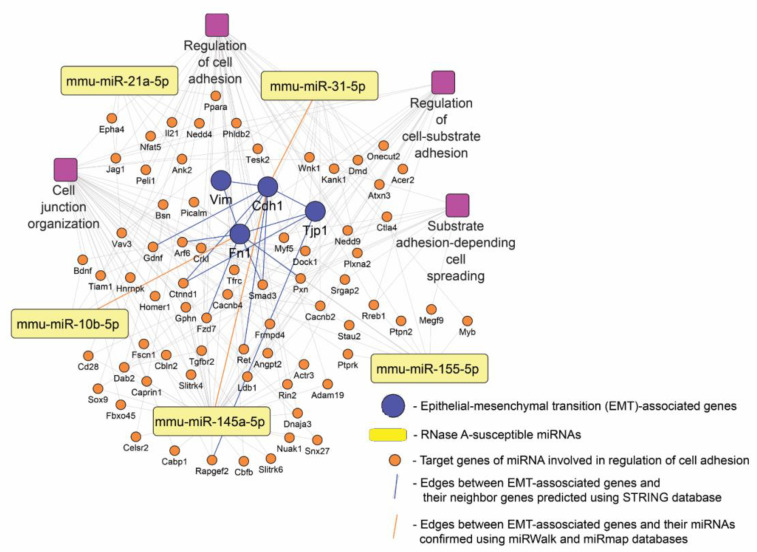
The probable regulatory network activated by RNase A in lung tissue adjacent to metastatic foci of mice with B16. The network was reconstructed using STRING database and visualized with Cytoscape 3.9.0. Adhesion-related pathways are indicated by purple squares.

**Figure 10 pharmaceutics-14-01173-f010:**
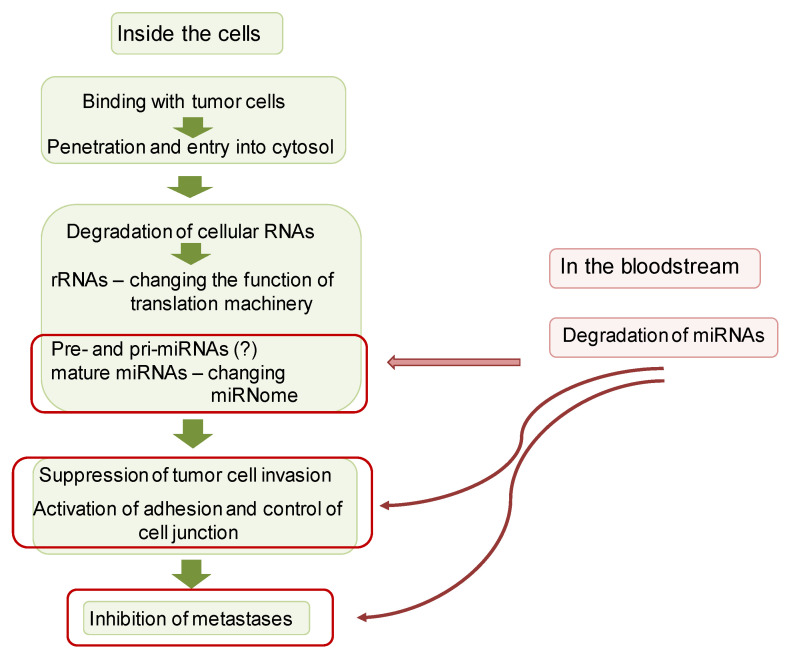
Hypothetical mechanism of antitumor activity of RNase A.

**Table 1 pharmaceutics-14-01173-t001:** The levels of miRNAs in B16 and HeLa cells after treatment with RNase A in vitro.

miRNA	Cells *w/o* Any Treatment		Cells Treated with RNase A	
	Absolute Expression Level, a.u. ^(2)^		Fold Changes in miRNA Level ^(3)^	
	B16	HeLa	B16	HeLa
mir-21a	0.7	0.9	n.a.	n.a.
mir-145a	0.8	0.95	n.a.	n.a.
mir-31	0.75	0.86	n.a.	n.a.
mir-10b	0.8	0.78	1.1↓–1.5↓	n.a.
let-7g	0.75	0.8	n.a.	n.a.
miR-155	0.56	1.0	1.6–2.0↓↓	n.a.
U6 snRNA ^(1)^	0.86	0.82	n.a.	n.a.

B16 and HeLa cells were incubated in the presence of RNase A (5 µg/mL) for 48 h under standard conditions. The levels of miRNA were determined by RT-qPCR. ^(1)^ U6 snRNA was used as a reference for normalization of miRNA level; ^(2)^ PCR meanings were normalized to the meaning of the most expressed sample and presented as arbitrary units (a.u.); ^(3)^ miRNA levels normalized to U6 snRNA. ↓ and ↓↓—decrease in the level of miRNA in 1.1–1.5 and 1.6–2.0 times, respectively; n.a.—not affected.

## Data Availability

Data are contained within the article or Appendix A.

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
