# Peer review of "Bovine Pancreatic RNase A: An Insight into the Mechanism of Antitumor Activity In Vitro and In Vivo"

_pharmaceutics, 2022, doi:10.3390/pharmaceutics14061173_

Round 1

Reviewer 1 Report

I thank editors for inviting me to review the submission numbered 1734156. This is a massive effort to unravel the mechanisms beyond antitumor action of Bovine pancreatic RNase A. In my opinion, the paper will be of interest for broad readership and will significantly contribute to the field. Therefore, I recommend publication of the manuscript after addressing the below points:

1- Too many papers have been cited in the introduction, more specifically in the first paragraph there are 14 citations which can easily be reduced.

2- Page 1, lines 37-38: the statement needs supporting reference.

3- Have you measured the initial scratch length or width? Based on the figure 4(A &C) it appears the gap between the margins of the scratch has been measured (not length). So, please correct the text both in MM and results.

4- What were the specifications of transwell inserts including pore size? Please add this info to the text (page 3), so the experiment can be reproducible.

5- Is it really possible to run a clonogenic assay in 96-well plate? Or it is just a typo error? Please check it (page 3, line 136).

6- Page 5, line 251: what do you mean by “two series of daily injections”? it is a bit confusing. You mean the animals were injected twice a day? Or just once a day? Can you also please explain why did you collect blood samples from the retro-orbital sinus? Is there any scientific reason or just personal preference? Please also add further description about the euthanasia of the animals.     

7- My main concern is statistics. Based on what the authors mentioned, they have used parametric tests (Student’s t-test and one-way ANOVA) to analyze the data, but how about if the data is not normally distributed? In some cases, huge error bars suggest non-normal distribution of data, which nonparametric tests should be employed.  

8- In figure 1, you have only compared different concentrations of RNase A to the control using student t test (one by one comparison). However, in the text you are talking about dose dependent penetration of the enzyme into the cells. This requires multiple comparisons between different doses. So, if you have done multiple comparisons, please add the statistics to the figure.

9- Based on figure 3(C and D), RNase A at 1 µM concentration has already significantly reduced total RNA content. So, please correct the text accordingly (page 12, line 475).

10- In figure 4(B), you have not compared different time points together. If you have, please add the differences level to the figure, to demonstrate that Incubation of B16 cells in the presence of RNase A time dependently reduced migration.

11- Figure 7D: you have not compared 0.7R to 7R, so how do you claim that the effect is dose-dependent (page 17, line 689).  

12- According to the figure 5A, the animals were treated with the enzyme 5 day after tumour inoculation, but in the text, it is 4 days! Again, the end point in the figure 5a is day 16, but in the text is claimed to be day 14. please consider the inconsistencies.

Minor points:

1- page 1, line 23: it should be microenvironment rather than environment.

2-   microRNA should first be abbreviated (in page 2, line 56) as miRNA and then be used in abbreviated format throughout the text.

3- Figure 2 legend (line 436): It should be B panel not D. 

4- Figure S2: if *p<0.05, **p<0.01, ***p<0.001, so what does this sign # mean?

5- Page 13, line 547: It should be DMEM not IMDM, please correct.

6- Page 15, line 615: it should be than, not that, please correct.  

7- CPPs stands for what? Please use full term if possible (page 20, line 769).

8- Page 15, line 624: please correct the sentence: “…with the level with the level…”

Author Response

1- Too many papers have been cited in the introduction, more specifically in the first paragraph there are 14 citations which can easily be reduced.

We reduced the number of citations. Please, see paragraph 1, marked by red.

2- Page 1, lines 37-38: the statement needs supporting reference.

Reference was added, marked by red.

3- Have you measured the initial scratch length or width? Based on the figure 4(A &C) it appears the gap between the margins of the scratch has been measured (not length). So, please correct the text both in MM and results.

Corrected. Please, see Section 2.3 of Materials and Methods, page 3, lines 125-126, marked by red.

4- What were the specifications of transwell inserts including pore size? Please add this info to the text (page 3), so the experiment can be reproducible.

Pore sizes for 16-CIM plates were introduced in the manuscript. Please, see Section 2.4.of Materials in methods, page 3, lines 133 and 140, marked by red.

5- Is it really possible to run a clonogenic assay in 96-well plate? Or it is just a typo error? Please check it (page 3, line 136).

There is no mistake. We used 96-well plates because we needed a small volume of medium and an area to seed only 200 cells as recommended in the protocol of clonogenic assay.

6- Page 5, line 251: what do you mean by “two series of daily injections”? it is a bit confusing. You mean the animals were injected twice a day? Or just once a day? Can you also please explain why did you collect blood samples from the retro-orbital sinus? Is there any scientific reason or just personal preference? Please also add further description about the euthanasia of the animals.     

The animals were injected once a day, we edited this information. Please, see Section 2.10 of Materials and Methods, page 6, lines 274, 276 - 277, marked by red.

Blood samples were collected from the retro-orbital sinus which is venous plexus located behind the eyeball. Thus, by introducing a capillary into a vein, we obtain venous blood that does not contact with air and, as a result, clots are formed more slowly compared to arterial blood obtained from the tail.

7- My main concern is statistics. Based on what the authors mentioned, they have used parametric tests (Student’s t-test and one-way ANOVA) to analyze the data, but how about if the data is not normally distributed? In some cases, huge error bars suggest non-normal distribution of data, which nonparametric tests should be employed.  

We used one-way ANOVA in the case of surface metastases and MII analysis. In spite of one-way ANOVA was reported to be suitable for both parametric (score data) and non-parametric (ranking/ordering) data before the analysis, we discarded the outliners and extreme values and brought the data to parametric distribution. We modified Figure 5, B, and added corresponding information to Figure 5 legend, page 17, lines 598, 603 – 604, marked by red.

8- In figure 1, you have only compared different concentrations of RNase A to the control using student t test (one by one comparison). However, in the text you are talking about dose dependent penetration of the enzyme into the cells. This requires multiple comparisons between different doses. So, if you have done multiple comparisons, please add the statistics to the figure.

Thank you very much for your remarks. We re-analyzed our cytometry data using one-way ANOVA, modified Figure 1, E-H, and added information on the performed analysis to Figure 1 legend. Please, see page 9, lines 419, 421-422, marked by red.

9- Based on figure 3(C and D), RNase A at 1 µM concentration has already significantly reduced total RNA content. So, please correct the text accordingly (page 12, line 475).

Corrected, page 12, lines 509, marked by red.

10- In figure 4(B), you have not compared different time points together. If you have, please add the differences level to the figure, to demonstrate that Incubation of B16 cells in the presence of RNase A time dependently reduced migration.

We did not compare different time points together. Intertemporal comparison in this case is impossible because we are forced to refer to the control specific to this particular time point.

11- Figure 7D: you have not compared 0.7R to 7R, so how do you claim that the effect is dose-dependent (page 17, line 689).

Corrected. Please, see page 18.

12- According to the figure 5A, the animals were treated with the enzyme 5 day after tumour inoculation, but in the text, it is 4 days! Again, the end point in the figure 5a is day 16, but in the text is claimed to be day 14. please consider the inconsistencies.

Thank you for careful consideration of our manuscript. Really, treatment was initiated on the day 4th after tumor implantation. 4, 5, 6, 7, 8, 11, 12, 13, 14, 15 – days of treatment once a day. On the day 15th of tumor growth 1 h after injection animals were sacrificed. All modifications were introduced in Figure 5 and in Figure legend. Please, see page 15, line 599, marked by red.

Minor points:

1- page 1, line 23: it should be microenvironment rather than environment.

Corrected.

2-   microRNA should first be abbreviated (in page 2, line 56) as miRNA and then be used in abbreviated format throughout the text.

Corrected

3- Figure 2 legend (line 436): It should be B panel not D. 

Corrected.

4- Figure S2: if *p<0.05, **p<0.01, ***p<0.001, so what does this sign # mean?

Corrected.

5- Page 13, line 547: It should be DMEM not IMDM, please correct.

Corrected.

6- Page 15, line 615: it should be than, not that, please correct.  

Corrected.

7- CPPs stands for what? Please use full term if possible (page 20, line 769).

Full term was introduced in the manuscript.

8- Page 15, line 624: please correct the sentence: “…with the level with the level…”

Corrected.

Reviewer 2 Report

In this paper, the authors reported the mechanism of antitumor activity of bovine pancreatic RNase A in vivo and in vitro. The experiments and results were well designed. However, I have some curious points in this report. Therefore, this paper is suitable for publishing at this stage.

Major:

  1. line 67, page 2, Why the author used two very different cell lines (B16 and HeLa cells) in this study? The explanation for choosing tumor cells was unclear and missing in this paper.
  2. Figure 1 G and H, the RFU values of RNase A in B16 and Hela cells were extremely low, indicating the fluorescent intensity of RNase A–biotin conjugate can be easily influenced by background signal or other noise. In addition, combined with the equation FI (RFU)=FIsample (RFU)-FIcontrol (RFU), it made me think the RFU between experiment and control groups were very closed.
  3. Why authors chose the Ku70/Ku80 heterodimer for molecular docking and co-localization? There might be many Ku dimers and why is Ku70/Ku80 heterodimer. If possible, please show more evidence.
  4. The author used docking and co-localization of fluorescence to reveal the interaction between Ku70/Ku80 and RNase A. But docking was computational and IF was not reliability at some times, the author should provide more evidence to verify the interaction between Ku70/Ku80 and RNase A. If the interaction was covalent, LC-MS/MS might be more convincing.
  5. Figure 3, the authors should check the representative picture. For example, 20 μM RNase A leaded to 80% decrease of total RNA content in B16 cells for 48 h, but the fluorescence of these two images had no obviously changes in my view.
  6. The effect of RNase A on adhesion and colony formation of B16 and HeLa cells was unreliability in Figure S2. The authors labeled significance differences between each column in Figure S2. However, there seems no significant difference in many groups.

Minor:

  1. line 28, page 1, the abbreviation of RI in the Keywords should be deleted and added at the first appearance in the Abstract.
  2. Please added the Em and Ex wavelength of confocal imaging in the Method.
  3. The number of E-H in Figure 1 legend was wrong.
  4. Please check the symbol of significance in Figure S2 legend. The symbol in figure and figure legend was inconformity.
  5. Figure 5, RNase A (0.7) and RNase A (7) were suggested to replace 0.7R and 7R in figure and figure legend.

Author Response

Major:

  1. line 67, page 2, Why the author used two very different cell lines (B16 and HeLa cells) in this study? The explanation for choosing tumor cells was unclear and missing in this paper.

We tried to investigate the effect of RNase A on cells of completely different histogenesis and origin (human and mouse). Mouse B16 melanoma is a melanocytic histogenesis cell line. Epidermoid carcinoma of the human cervix HeLa is a cell line of epithelial origin. Both lines, despite their differences in histogenesis and origin, form a monolayer, which is a mandatory criterion for studying migration, mobility, and invasion.

We added our criteria for choosing cell line in the manuscript. Please, see page 8, lines 370 - 373, marked by red.

  1. Figure 1 G and H, the RFU values of RNase A in B16 and Hela cells were extremely low, indicating the fluorescent intensity of RNase A–biotin conjugate can be easily influenced by background signal or other noise. In addition, combined with the equation FI (RFU)=FIsample (RFU)-FIcontrol (RFU), it made me think the RFU between experiment and control groups were very closed.

We are very grateful to you for careful reading of the manuscript. Indeed, in Figure 1 we have shown the data without subtracting the control fluorescence from the sample fluorescence. We introduced corrections in Materials and Methods. Please, see section 2.8, page 5.

The level of fluorescence intensity in B16 cells was really low at the concentration of the conjugate RNase A – biotin 1 and 5 μM. And we observed no statistically reliable differences between these groups and control not only in terms of fluorescence intensity but also in terms of number of fluorescence cells. The same can be said for HeLa cells at the same doses of the conjugate.

We introduced this information in the manuscript. Please, see page 8, lines 393 - 398, marked by red.

  1. Why authors chose the Ku70/Ku80 heterodimer for molecular docking and co-localization? There might be many Ku dimers and why is Ku70/Ku80 heterodimer. If possible, please show more evidence.

Ku is one of the cellular proteins executing multiple functions whose intracellular concentration is amounted to 1.5 μM [6]. Ku is a complex composed of two tightly associated subunits called Ku70 and Ku80 [1] and localized both in cytosol and nucleus. Ku has long been considered as a nuclear protein playing a role in DNA repair [2], but there are now some evidence of Ku functions in cytosol and on outer membrane: cytosolic DNA sensing and subsequent innate immune response activation [9], apoptosis regulation, mitosis [10], hypoxia, metabolism, and inflammatory response [11]. Ku can recognize RNA hairpins although less effectively than dsDNA [7] so it is considered as RNA-binding protein.

Recent reports demonstrate that some mammalian cells are using the DNA repair protein Ku in a membrane-associated form for interaction with their microenvironment composed of other cellular components and extracellular matrix (ECM) [3]. The functions of the membrane associated Ku are not restricted to cell-cell interaction but also include cell-matrix-interaction. Additional results, although mainly descriptive, raised the idea that Ku might also play a role in cell migration and invasion. On the other hand, several studies have revealed the cytoplasmic and the cell-surface localization of Ku proteins in a variety of tumor cells, including leukemia, multiple myeloma and solid tumor cell lines [4].

Taking into account the high abundance of the Ku protein in the tumor cells and its participation in cell contacts and invasion, it seemed interesting to evaluate its interaction with RNase A at the level of the tumor cell.

We introduced this information in the Introduction. Please, see Introduction, page 6, lines 67 – 86, marked by red.

  1. The author used docking and co-localization of fluorescence to reveal the interaction between Ku70/Ku80 and RNase A. But docking was computational and IF was not reliability at some times, the author should provide more evidence to verify the interaction between Ku70/Ku80 and RNase A. If the interaction was covalent, LC-MS/MS might be more convincing.

Indeed, despite revealed low Gibbs free energy and Kd (Fig. 2A), molecular modeling can only show the probability of direct interaction of RNase A and Ku70/Ku80 heterodimer and identify their probable binding sites. Further confocal microscopy demonstrated co-localization of fluorescent signals related to RNase A and Ku70/Ku80 in tumor cells (Fig. 2C,D) that can indicate protein-protein interaction. In order to more thoroughly explore RNase A/Ku70/Ku80 binding more subtle approaches, such as immunoprecipitation (IP) or surface plasmon resonance, should be used. However, we should note that our study was aimed to evaluate probable mechanisms of RNase A intracellular trafficking; and more comprehensive investigation of direct interaction of RNase A with Ku proteins using IP or physical methods will be the objective of our further study.

  1. Figure 3, the authors should check the representative picture. For example, 20 μM RNase A leaded to 80% decrease of total RNA content in B16 cells for 48 h, but the fluorescence of these two images had no obviously changes in my view.

Fluorescence was assessed not visually, which is very difficult to do in the case of staining of cells with a SYTO, but quantitatively. The fluorescence intensities of intracellular RNA were measured in the sample images obtained with confocal fluorescent microscopy on LSM710 (Zeiss, Germany) by using ImageJ software. Fluorescence intensity was calculated using the equation:

Fluorescence intensity, % = (RFUexperimental group/RFUcontrol group) ×100%, where RFUcontrol group is the fluorescence intensity of non-treated cells, and RFUexperimental group is the fluorescence intensity of cells treated with RNase A.

This information is presented in section 2.6. of Materials and Methods, page 4. We also introduced some clarifications in the manuscript. Please, see page 12, lines 507  508, marked by red.

  1. The effect of RNase A on adhesion and colony formation of B16 and HeLa cells was unreliability in Figure S2. The authors labeled significance differences between each column in Figure S2. However, there seems no significant difference in many groups.

Really, at the concentrations of RNase A 0.625, 1.2 and 2.5 μg/ml we did not observe any statistically significant effects of RNase A on adhesion and colony formation (on Figure S2 “#” is presented but in Figure’s S2 legend we did not specify that this icon “#” indicates statistically insignificant difference). We modified Figure S2 and clarified this issue in the manuscript.

Minor:

  1. line 28, page 1, the abbreviation of RI in the Keywords should be deleted and added at the first appearance in the Abstract.

Corrected.

  1. Please added the Em and Ex wavelength of confocal imaging in the Method.

Information was introduced in Sections 2.6, 2.7 and 2.8 of Materials and Methods. Marked by red.

  1. The number of E-H in Figure 1 legend was wrong.

Corrected.

  1. Please check the symbol of significance in Figure S2 legend. The symbol in figure and figure legend was inconformity.

Corrected.

  1. Figure 5, RNase A (0.7) and RNase A (7) were suggested to replace 0.7R and 7R in figure and figure legend.

We modified Figures 5 – 7 and modified text according to your recommendations.

Round 2

Reviewer 1 Report

The paper has nicely been revised, but please consider the below point:

-  I looked into the paper by Guzman, et al 2014. (Colony Area: an ImageJ plugin to automatically quantify colony formation in clonogenic assays. PloS one, 9(3), p.e92444). Based on their pictures and M&M, they have planted the cells in 12-well plates. It is still a question for me, how it is possible to run colonogenic assay in a 96-well plate and count the colonies.

Author Response

-  I looked into the paper by Guzman, et al 2014. (Colony Area: an ImageJ plugin to automatically quantify colony formation in clonogenic assays. PloS one, 9(3), p.e92444). Based on their pictures and M&M, they have planted the cells in 12-well plates. It is still a question for me, how it is possible to run colonogenic assay in a 96-well plate and count the colonies.

Colony formation assay was performed using a methodology adapted for the 96-well plate. The number of cells per well was reduced and the incubation time was increased. This method was first used in our laboratory, the data are published in Markov et al, Molecules, 2020, 25(24):5925.

We introduced clarification to the text and replaced the reference. Please, see section 2.4, page 4, lines 158 – 159, marked by red.

Reviewer 2 Report

The authors has addressed my questions.

Author Response

We are very grateful to you for carefully studying our article. Your comments have greatly improved the manuscript.